# Unidirectional MCM translocation away from ORC drives origin licensing

Agata Butryn ⓘ, Julia F. Greiwe & Alessandro Costa ⓘ ✉

The MCM motor of the eukaryotic replicative helicase is loaded as a double hexamer onto DNA by the Origin Recognition Complex (ORC), Cdc6, and Cdt1. ATP binding supports formation of the ORC-Cdc6-Cdt1-MCM (OCCM) helicase-recruitment complex where ORC-Cdc6 and one MCM hexamer form two juxtaposed rings around duplex DNA. ATP hydrolysis by MCM completes MCM loading but the mechanism is unknown. Here, we used cryo-EM to characterise helicase loading with ATPase-dead Arginine Finger variants of the six MCM subunits. We report the structure of two MCM complexes with different DNA grips, stalled as they mature to loaded MCM. The Mcm2 Arginine Finger-variant stabilises DNA binding by Mcm2 away from ORC/Cdc6. The Arginine Finger-variant of the neighbouring Mcm5 subunit stabilises DNA engagement by Mcm5 downstream of the Mcm2 binding site. Cdc6 and Orc1 progressively disengage from ORC as MCM translocates along DNA. We observe that duplex DNA translocation by MCM involves a set of leading-strand contacts by the pre-sensor 1 ATPase hairpins and lagging-strand contacts by the helix-2-insert hairpins. Mutating any of the MCM residues involved impairs high-salt resistant DNA binding in vitro and double-hexamer formation assessed by electron microscopy. Thus, ATPase-powered duplex DNA translocation away from ORC underlies MCM loading.

Loading and activation of the MCM replicative helicase are separated in time, to ensure that any given DNA segment in the eukaryotic genome is duplicated only once per cell cycle[1,2]. MCM is a ring-shaped helicase containing six related AAA+ ATPase polypeptides featuring bipartite active sites with catalytic residues contributed by neighbouring subunits[3,4]. In *Saccharomyces cerevisiae*, MCM is bound by loading factor Cdt1[5,6], which keeps the ATPase ring in a state competent for loading onto DNA. It stabilises a discontinuity within the Mcm2 and Mcm5 subunits of the MCM ring, which forms a gate through which DNA is threaded ("DNA gate")[7–10]. In late mitosis and throughout G1 phase, origins are licensed for replication. During this process, MCM is loaded onto DNA as a head-to-head double hexamer (DH) that contains the symmetry to support bidirectional replication but remains catalytically inactive[2,5,11]. Activation occurs upon S-phase transition when Cdc45 and Go-Ichi-Ni-San (GINS) bind MCM to form

the CMG complex[12–14]. CMG melts the double helix[15], ejects the lagging-strand template[16,17] and unidirectionally translocates along the leading-strand template with 3′ to 5′ polarity to unwind the established replication fork[13].

Work with yeast proteins revealed that DH formation primarily occurs via a sequential mechanism where loading of a first single MCM hexamer (SH) is required for second SH recruitment[18,19]. Loading of both hexamers[20] requires a short-lived ORC-Cdc6-Cdt1-MCM (OCCM) intermediate that can be stabilised by using the slowly hydrolysable ATP analogue, ATPγS[21,22]. On the path to OCCM formation, ORC clamps around and bends DNA[23], recruits Cdc6 to form a ring that fully encircles DNA[24,25], and then recruits MCM-Cdt1 that recognises ORC-Cdc6 initially via the C-terminal Winged Helix Domains (WHD)s of Mcm3 and Mcm7. In this ORC-Cdc6-MCM-Cdt1 (OC-MC[18]) intermediate, also known as pre-insertion OCCM[26], the DNA bent by ORC is

Macromolecular Machines Laboratory, The Francis Crick Institute, London NW1 1AT, UK. ²Present address: Astex Pharmaceuticals, 436 Cambridge Science Park Milton Rd, Milton, Cambridge CB4 0QA, UK. ✉e-mail: alessandro.costa@crick.ac.uk

aligned with the DNA gate, ready to be threaded through the MCM central channel. DNA then straightens and enters the MCM pore, achieving OCCM formation. The MCM ring in OCCM is notched, with the Mcm2-5 gate closed only via an alpha-helical latch of Mcm2 that engages the neighbouring Mcm5 ATPase domain[22]. Release of MCM from ORC-Cdc6 results in the formation of the first MCM ring that is topologically loaded around DNA and features a completely closed DNA gate[27]. At this stage, ORC disengages DNA and switches from binding the C- to the N-terminal side of a first loaded SH, forming the MCM-ORC (MO) intermediate[18,28]. In this configuration, ORC is competent for recruiting a second MCM-Cdt1 complex via the same mechanism as the first[20]. The result is an MCM-ORC-Cdc6-MCM-Cdt1 intermediate (MOC-MC, i.e. the second pre-insertion OCCM), which is later converted to fully-loaded DH[18] (Fig. 1).

Biochemical studies established that ATP hydrolysis by MCM, but not ORC or Cdc6, are essential to complete helicase loading. In fact, MCM variants containing changes in the Walker A (ATP-binding) or Arginine Finger (ATP-hydrolysis) motifs of any of the six MCM subunits are defective for loading[20,29,30]. Despite this knowledge, it is unclear how ATP hydrolysis by MCM promotes DH loading. To address this issue, we took cryo-electron microscopy (cryo-EM) approach to analyse DH loading reactions using inactive ATPase variants of the six MCM subunits.

## Results

### MCM loading with six MCM Arginine Finger mutants

To understand the role of ATP hydrolysis by MCM during DH loading, we used negative stain electron microscopy to test the effect of alanine substitution of Arginine Finger (RA mutants) of Mcm2 (R676A), 3 (R542A), 4 (R701A), 5 (R549A), 6 (R708A) or 7 (R593A) in a DH loading reaction with purified yeast proteins[29] (Supplementary Fig. 1). The DNA template used contains the well-characterised yeast origin sequence, ARS1, capped at both ends by protein roadblocks that prevent MCM from sliding off DNA (an HpaII methyltransferase, MH, followed by a nucleosome at the 5′ end and a second nucleosome at the 3′ end[18]). As previously observed with DNA-affinity purification using streptavidin-coated magnetic beads[29,30], all MCM variants were defective for DH formation, although to different degrees depending on what subunit contained the RA change. Mcm4RA gave the mildest defect, with a drop in DH count from $56.5 \pm 2.7\%$ of averageable MCM particles (wild type) to $22.4 \pm 2.9\%$. Progressively more pronounced defects were observed for Mcm3RA ($13.9 \pm 5.6\%$), Mcm7RA ($5.3 \pm 1.6\%$), Mcm6RA ($2.4 \pm 0.1\%$), Mcm2RA ($0.5 \pm 0.4\%$) and Mcm5RA (no DH, Fig. 2). The DH loading defects observed by EM for each mutant strikingly match results from bead-based loading assays conducted in two separate studies[30,31].

As the EM analysis was performed on the entire loading reaction in solution and with no downstream purification, we could also observe all known DH loading intermediates, otherwise lost with the stringent washing conditions of the bead-based assay, generally used to isolate topologically loaded complexes[30,31]. Different intermediates were enriched using different ATPase variants, allowing us to establish during what step in the DH loading reaction fails. Mcm3RA, Mcm4RA, Mcm6RA and Mcm7RA, which still yield sizeable fractions of loaded DHs, were enriched for OC-MC, MO and MOC-MC intermediates. This indicates a defect in the threading of duplex DNA through the Mcm2-5 gate for the ATP-hydrolysis-powered reaction, but not in OCCM

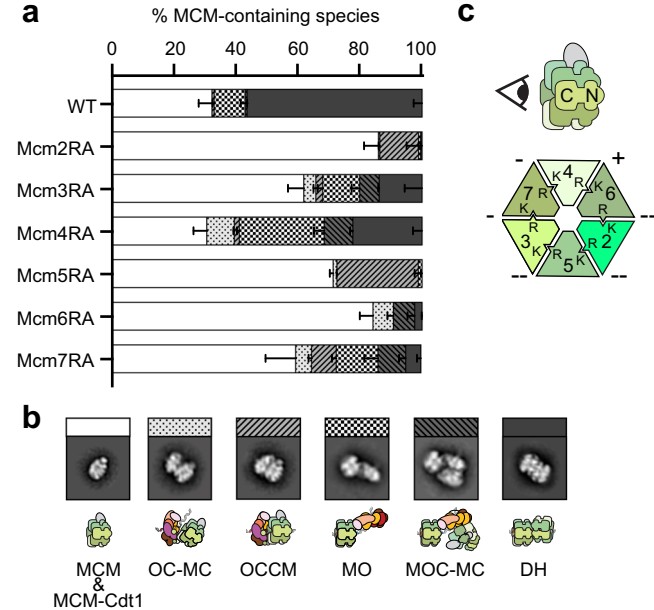

**Fig. 2 | Quantification of DH loading on a roadblocked ARS1 origin template using MCM RA mutants. a** DH formation was assessed by negative stain analysis and 2D averaging. Data represent mean ± SD. N = 3 independent experiments. Source data are provided as a Source Data file. **b** Representative 2D averages and cartoon models corresponding to detected MCM loading intermediates. **c** Schematic representation of the relative effects of RA mutations on the efficiency of DH formation around the MCM ring view from the C-terminus. The percent of activity retained after mutation is shown next to the targeted ATPase sites, with 0–5% of WT indicated by "--"; 5–15%, "-"; and 15–25%, "+". OC-MC ORC-Cdc6-MCM-Cdt1 complex, OCCM ORC-Cdc6-Cdt1-MCM complex, MO MCM-ORC complex, MOC-MC MCM-ORC-Cdc6-MCM-Cdt1 complex, DH MCM double hexamer.

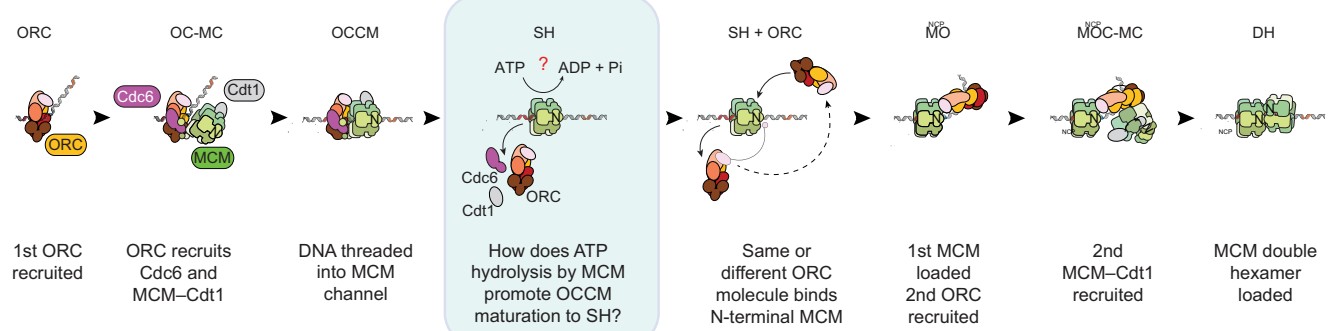

**Fig. 1 | Sequential mechanism for double hexameric MCM loading onto origin DNA.** A key unresolved question addressed in this study is the essential role of ATP hydrolysis by MCM during loading. OC-MC: ORC-Cdc6-MCM-Cdt1 complex.

OCCM: ORC-Cdc6-Cdt1-MCM complex. SH single-loaded MCM hexamer, MO MCM-ORC complex, MOC-MC MCM-ORC-Cdc6-MCM-Cdt1 complex, DH MCM double hexamer.

maturation. Conversely, Mcm2RA and Mcm5RA yielded barely detectable or no loaded DHs and were instead enriched for OCCM-like intermediates (Fig. 2). This indicates that introducing RA mutations in either of the two DNA gate subunits of MCM blocks maturation of OCCM to SH.

## Cryo-EM analysis of DH loading with the Mcm2RA variant

To better understand the role of ATP hydrolysis during OCCM to SH transition, we employed cryo-EM to analyse the DH loading reaction on MH-capped ARS1 DNA, using MCM-Cdt1 complexes containing the Mcm2RA variant (Supplementary Fig. 2a). Two-dimensional (2D) classification in RELION revealed three categories of particles: free MCM complexes (most abundant), ATPγS-OCCM like particles[22] and DHs (least abundant, Supplementary Fig. 2b). We focused on particles that resembled OCCM (hereafter OCCM$^{Mcm2RA}$), which we further subjected to three-dimensional (3D) reconstruction. We solved a 3.3 Å resolution consensus structure and separate local refinements of the ORC-Cdc6 and MCM-Cdt1 regions of the map improved the resolution to 3.2 Å for each sub-volume (Supplementary Fig. 2c–f, Supplementary Fig. 3 and Supplementary Table 1). Cryo-EM density for all subunits of the ATPγS-OCCM complex could be recognised in OCCM$^{Mcm2RA}$, however, the topologically closed ORC-Cdc6 ring transitions to a notched state, while the notched MCM ring becomes topologically closed. Moreover, cryo-EM density for Cdc6 in OCCM$^{Mcm2RA}$ becomes fragmented, reflecting increased flexibility. This density could be best recovered, though only to nanometre resolution, with deep-learning-based map improvement using EMReady[32] (Fig. 3a). Density for the Cdc6-interacting Orc1 ATPase is low quality and the Orc2 WHD domain becomes disordered. As a result, duplex DNA appears merely clamped by ORC-Cdc6, and not fully encircled by a protein ring, as observed in ATPγS-OCCM[22] (Fig. 3a). Two tethering elements that join the ORC-Cdc6 with the MCM ring also become flexible. One is Orc6, for which only two C-terminal alpha helices could be built (Fig. 3a), different from ATPγS-OCCM, where the entire C-terminal cyclin box could be observed, nestled between Orc3 and the Mcm2 ATPase domain[22]. The second one is the Mcm3 WHD, which docks against Cdc6 and Orc2 WHD in ATPγS-OCCM[22], but becomes disordered in OCCM$^{Mcm2RA}$.

When comparing the MCM assembly in ATPγS-OCCM and OCCM$^{Mcm2RA}$, several conformational changes can be detected. The largest-scale transition is the closure of the Mcm2-5 gate (Fig. 3b). The N-terminal tier of the MCM ring also becomes topologically closed. The Mcm2RA N-terminal B domain transitions from partially plugging the MCM central channel (ATPγS-OCCM state) to a retracted configuration that creates the space for full access of the double helix into the MCM pore (Fig. 3c). In fact, continuous cryo-EM density for duplex DNA can be observed spanning the entire length of the MCM ring, unlike ATPγS-OCCM where the double helix is only seen encircled by the ATPase tier (Fig. 3d).

The subunit arrangement and the protein-DNA contacts differ in OCCM$^{Mcm2RA}$ compared to ATPγS-OCCM[22]. In the ATPγS-stabilised structure, the ATPase subunits of Mcm7, Mcm4, Mcm6 and Mcm2 are arranged in a nucleotide-stabilised, tightly interacting right-handed spiral arrangement that follows the helicity of DNA. Pre-sensor 1 (PS1) hairpin lysines of Mcm7, 4 and 6 project from the ATPase domain and bind selectively the leading-strand template[22] (Fig. 4a). In OCCM$^{Mcm2RA}$, Mcm7 disengages from the ATPase spiral and from DNA, while Mcm2 forms new contacts with the double helix. As observed for the other DNA-engaged MCM subunits, the PS1 hairpin of Mcm2 selectively binds the leading-strand template. Because a longer DNA segment occupies the MCM pore in OCCM$^{Mcm2RA}$, one additional set of interactions can be observed, with the helix 2 insert (h2i) hairpin of Mcm4, 6 and 2 engaged in a helical pattern and predominantly binding the lagging-strand template (Fig. 4b).

Differences in nucleotide occupancy can also be observed. In ATPγS-OCCM, four ORC sites are occupied by ATPγS (Orc1-Cdc6,

Orc4-Orc1, Orc5-Orc4 and Orc3-Orc5)[22]. In OCCM$^{Mcm2RA}$, instead, three sites were ATP bound, while no nucleotide density could be resolved in the Orc1-Cdc6 active site (Fig. 5a). Within the MCM ATPase, four sites are occupied in ATPγS-OCCM, while all active sites are nucleotide-bound in the six MCM subunits of OCCM$^{Mcm2RA}$ (Fig. 3b). We assigned them as ADP based on the cryo-EM density (Supplementary Fig. 4).

## Cryo-EM analysis of DH loading with the Mcm5RA variant

To further understand the role of ATP hydrolysis in OCCM-to-SH transition, we assembled a DH loading reaction using MCM-Cdt1 bearing the Mcm5RA variant (Supplementary Fig. 5a). We ignored non-DNA loaded MCM complexes and analysed particles superficially reminiscent of OCCM (Supplementary Fig. 5b). 3D classification identified two distinct states of a complex containing ORC-Cdt1-MCM but lacking Cdc6 (OC$_1$M$^{Mcm5RA}$, Supplementary Fig. 6). While all ORC subunits are visible, the two states differ in the configuration of Orc1. In conformer 1, solved to 3.7 Å resolution (Supplementary Fig. 5c, d and Supplementary Table 2) both the C-terminal WHD domain of Orc1 can be observed as well as the ATPase domain (though local resolution of the latter is lower, Supplementary Fig. 5d). Orc1 ATPase is engaged in AAA+ interaction with the neighbouring Orc4 subunit, with ATP bound at the interface (Fig. 5a, Supplementary Fig. 4). In conformer 2, solved to 3.4 Å resolution in the consensus structure (or 3.3 and 3.5 Å when locally refining ORC and MCM-Cdt1, respectively, Supplementary Fig. 5e–g, Supplementary Fig. 7, Supplementary Table 2), only the WHD but not the ATPase domain of Orc1 is visible. As a result, only two ATP molecules could be built in the ORC complex (at the Orc5-Orc4 and Orc3-Orc5 interfaces, Fig. 5a, Supplementary Fig. 4). The connectivity between ORC and MCM also changes. One new contact is formed by Mcm5 WHD (unstructured in OCCM$^{Mcm2RA}$) that engages Orc3 WHD. One contact is lost, with Mcm7 WHD, which touches the Orc1 ATPase in OCCM$^{Mcm2RA}$ and ATPγS-OCCM[22], becoming disordered in OC$_1$M$^{Mcm5RA}$ (Fig. 5b). Other ORC-MCM contacts observed in ATPγS-OCCM that are broken in OCCM$^{Mcm2RA}$ are also absent in OC$_1$M$^{Mcm5RA}$.

Structural changes can be detected within the MCM ring. Compared to the OCCM$^{Mcm2RA}$, which contains a three-subunit ATPase spiral (Fig. 4b), the Mcm5 subunit engages Mcm2 in OC$_1$M$^{Mcm5RA}$, capping the N-terminal end of a four-subunit spiral (Fig. 4c). Nucleotide (ADP, according to the cryo-EM density, Supplementary Fig. 4) can be observed bound to each of the Mcm4-6-2-5 subunits forming the ATPase spiral, but not to the Mcm3 and Mcm7 subunits (Fig. 5c). Mcm5 establishes new contacts with DNA downstream of the Mcm2 DNA binding sites and away from ORC. As observed for all other DNA-bound MCM subunits, the Mcm5 PS1 hairpin selectively engages the leading strand downstream of Mcm2 and the h2i hairpin predominantly binds the lagging strand (Fig. 4c).

## A role for DNA-interacting ATPase hairpins in MCM loading

We observed that blocking OCCM-to-SH transition by using the Mcm2RA and Mcm5RA variants changes the mode in which DNA is gripped by the MCM ring, with different ATPase subunits engaging the double helix through PS1 and h2i hairpin residues in the two mutant structures, compared to the ATPγS-OCCM structure[22] (Supplementary Fig. 8). We, therefore, asked whether these different DNA contacts are important for MCM loading. We designed Mcm7,4,6,2,5 PS1 and h2i hairpin mutations based on the published OCCM[22] and OC$_1$M$^{Mcm5RA}$ structure presented in this work, and one Mcm3 PS1 hairpin mutation, based on sequence alignment and the recently published structure of a loaded SH[27] (Supplementary Fig. 9a–c). No equivalent positively charged residue in Mcm3 h2i could be identified contacting DNA. To test the importance of these protein-DNA interactions in MCM loading, we used a variation of the previously published DNA-affinity bead-based assay that can discriminate between ATPγS-dependent MCM recruitment and ATP-hydrolysis-dependent loading[5]. As DNA bait we used the ARS1 origin sequence capped at both ends with a protein

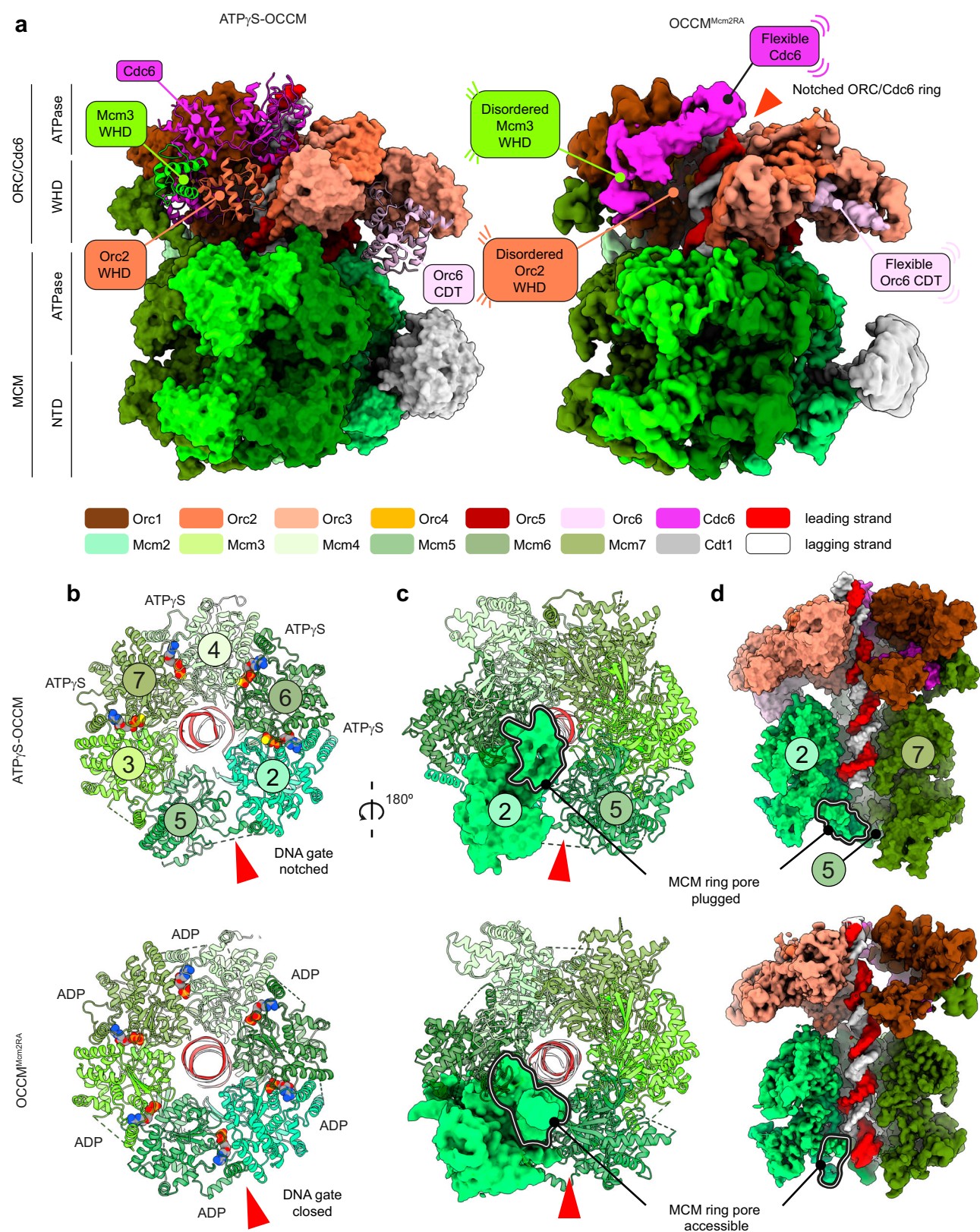

roadblock, which prevents MCM from sliding off DNA (in this case, Trp repressor). We used Strep-Tactin beads to pull on a Twin-Strep-tag fused to the Trp repressor, to isolate MCM recruited or loaded onto origin DNA. Lysine to alanine (KA) changes in the PS1 hairpin of any of the six ATPase subunits that contact the leading-strand template (Mcm2$^{K633A}$, Mcm3$^{K499A}$, Mcm4$^{K658A}$, Mcm5$^{K506A}$, Mcm6$^{K665A}$ Mcm7$^{K550A}$) were all defective for MCM loading, with Mcm6 having the least severe

defect (Supplementary Fig. 9d). Under the same experimental conditions, no defect was detected instead when mutating positively charged residues observed to interact with the lagging strand in the h2i hairpin of OCCM and OC$_1$M complexes (Mcm2$^{K582A}$, Mcm4$^{R607A}$, Mcm5$^{R455A}$, Mcm6$^{R614A}$, Mcm7$^{K499A}$, Supplementary Fig. 9d). Inspection of our OCCM$^{Mcm2RA}$ and OC$_1$M$^{Mcm5RA}$ structures explains this initial observation. In fact, while each PS1 hairpin lysine engages in one

**Fig. 3 | Comparison between ATPγS-OCCM and OCCM$^{Mcm2RA}$ structures. a** Side views of the two OCCM structures highlight structural changes in ORC-Cdc6 and at the interface with MCM. In OCCM$^{Mcm2RA}$ (right, cryo-EM density) Cdc6 and the C-terminal cyclin box domain become flexible, and the Orc2 WHD domain becomes disordered, compared to ATPγS-OCCM (left, surface representation). The Mcm3 WHD domain that contacts Cdc6 and Orc1 also becomes disordered in OCCM$^{Mcm2RA}$. **b** MCM ATPase view shows that the Mcm2-5 gate is notched in ATPγS-OCCM (top) and closed in OCCM$^{Mcm2RA}$ (bottom). ATPγS is bound at four inter-subunit sites in ATPγS-OCCM. Instead, ADP is found engaged at all sites in OCCM$^{Mcm2RA}$. **c** N-terminal MCM view shows that the Mcm2-5 gate is notched in ATPγS-OCCM (top) and closed in OCCM$^{Mcm2RA}$ (bottom). The B-domain of Mcm2 that plugs the MCM pore in ATPγS-OCCM (top, surface representation) is reconfigured so that duplex DNA can be fully loaded in OCCM$^{Mcm2RA}$ (bottom, cryo-EM density). **d** Cut-through side view shows that duplex DNA traverses ORC-Cdc6 and the MCM ATPase in ATPγS-OCCM (top, surface representation), while it spans the entire length of the MCM hexamer in OCCM$^{Mcm2RA}$ (bottom, cryo-EM density).

hydrogen bond with every second phosphate of the leading strand, h2i hairpins engage in several contacts with both DNA strands, including backbone phosphate and sugar, as well as nitrogenous bases, promoting some widening of the minor groove (Supplementary Fig. 10).

Given the extensive nature of the h2i contacts, we asked whether more stringent biochemical conditions should be used to detect defects in DNA loading, when individual h2i residues are mutated. We found that all h2i hairpin mutants were defective for ATP-dependent loading but not for recruitment in ATPγS, when MCM-Cdt1 concentration was dropped from 550 to 220 nM and ATP (or ATPγS) dropped from 5 mM to 315 μM (Fig. 6a). Similar loading defects were obtained for the PS1 hairpin mutants tested under these stringent conditions (Fig. 6b). Individual Mcm2 and Mcm5 h2i hairpin mutants yielded low levels of high salt-stable binding, but a combination of these two mutants (Mcm2$^{K582A}$-Mcm5$^{R455A}$, 2A) was sufficient to completely abrogate any loading (Supplementary Fig. 9e). This indicates that the DNA contacts established in OCCM$^{Mcm2RA}$ and OC$_1$M$^{Mcm5RA}$, but absent in ATPγS-OCCM, are important for OCCM maturation to fully loaded MCM. Equally defective loading was observed using a variant that targets the h2i residues engaged to the lagging strand in all four ATPase spiral subunits of the OC$_1$M$^{Mcm5RA}$ complex (Mcm2$^{K582A}$-Mcm4$^{R607A}$-Mcm5$^{R455A}$-Mcm6$^{R614A}$, 4A, Fig. 6a and Supplementary Fig. 9e). Negative stain analysis of the full loading reaction supports our observations. In fact, DH loading is severely impaired by individual changes in Mcm4,6,2,5 for either the PS1 or h2i hairpin variants (Fig. 6c, Supplementary Fig. 11). Displaying the ratio between OCCM and DH assemblies in our loading reactions highlights the importance of both the Mcm2 PS1 and h2i hairpin interactions with DNA during OCCM maturation and MCM loading (Supplementary Fig. 9f). It also reinforces our observation of a role for the h2i hairpin contacts with the lagging-strand templates in MCM loading, given the strong loading defect observed of the 4 A mutant that targets h2i in Mcm4,6,2,5.

## Discussion

Three ATPase factors (ORC, Cdc6 and MCM) are involved in origin licensing, but the role of ATP binding and hydrolysis is not fully understood. ATP binding by both ORC and Cdc6 is essential for DH formation, but ATP hydrolysis is only required for the release of non-productive MCM loading intermediates and not for helicase loading itself[20,29]. Conversely, ATP binding and hydrolysis by certain MCM subunits are essential for DH loading[10,29,30]. By solving the structures of OCCM maturation intermediates stalled with two separate MCM ATPase mutations, we sought to uncouple the effect of ATP hydrolysis by ORC-Cdc6 from ATP hydrolysis by MCM. This approach, combined with the previous work on OCCM stalled with ATPγS, led to capturing three structures representing different compositional and conformational intermediates of the OCCM maturation path. We assigned an order for these loading reaction intermediates based on how complete or disrupted the ORC-Cdc6 ring was and on the gradual loss of the interactions between WHDs of Mcm3 and 7 and ORC-Cdc6. In our OCCM$^{Mcm2RA}$ structure, we observed that the Mcm3 WHD and Cdc6 become flexible and the ORC-Cdc6 ring becomes topologically open. Upon this transition, the Orc1-Cdc6 interface changes from a tight configuration that favours ATP binding (ATPγS-OCCM[22]) to a relaxed configuration that would favour product release after ATP hydrolysis.

This agrees with the observation that the Mcm3 WHD contact with Cdc6, established upon ATP binding in OCCM[22], stimulates ATP hydrolysis within the Orc1-Cdc6 catalytic site[20].

In turn, for OC$_1$M$^{Mcm5RA}$ we observe two states, co-existing after Cdc6 release. They differ from the tight interaction that favours ATP binding at the Orc4-Orc1 interface observed in ATPγS-OCCM[22] and inform us on structural flexibility that would promote product release after ATP hydrolysis at the same site. In one state, the full Orc1 subunit is visible, and its ATPase domain is flexible yet still engaged to the neighbouring Orc4 subunit. In the second state, only the C-terminal Orc1 WHD is detected, while the Orc1 ATPase is completely disengaged from Orc4 and becomes disordered, similar to the structure of ORC in higher eukaryotes[33,34]. Thus, as ORC and Cdc6 perform their catalytic function, they progressively break the topologically closed ring structure formed around DNA upon ATP binding[22,24,25]. They achieve this through the release of the Cdc6 ATPase subunit first, and then through disengagement of the Orc1 ATPase module from the rest of ORC (represented by conformer 1 and 2, respectively, Fig. 7a).

These three structures increase our understanding of the ATPase function by the ORC and Cdc6 loaders and provide insights into the release of incomplete MCM complexes. In fact, earlier biochemical observations established that, when Cdt1 is absent, ATPγS-ORC-Cdc6 only retains the Mcm5-3-7 subcomplex (incomplete MCM) but not Mcm4-6-2[20]. Mobilisation of the Cdc6 and Orc1 subunit that we see occurring in ATP-hydrolysis permissive conditions abrogates the interaction with Mcm3 and Mcm7, as observed in our OC$_1$M$^{Mcm5RA}$ structure). This structural change explains the release of the incomplete 5-3-7 MCM subcomplex triggered upon ATP hydrolysis by ORC/Cdc6[20] (Fig. 7a), providing the structural basis for the proofreading mechanism enacted by ORC/Cdc6 ATP hydrolysis during origin licensing.

Our structures of sequentially disassembling ORC/Cdc6 also suggest a temporal order for the maturation of OCCM to SH (ATPγS-OCCM → OCCM$^{Mcm2RA}$ → OC$_1$M$^{Mcm5RA}$ → SH). We note that the order of events derived from our structures agrees with single-molecule co-localisation observations of MCM loading, performed with wild-type or Arginine Finger mutant proteins[28,35,36]. These studies previously implicated ATP hydrolysis by MCM in facilitating sliding along DNA and showed that ORC release occurs after closure of the Mcm2-5 gate, in accord with our structural observations. The emerging order of events has implications for the mechanism of disengagement of loaded MCM from ORC. In fact, inspection of the ATPase tier in ATPγS-OCCM, OCCM$^{Mcm2RA}$ and OC$_1$M$^{Mcm5RA}$ suggests that, as ATP is hydrolysed in neighbouring catalytic sites, the PS1 and h2i hairpins of Mcm2 and Mcm5, which project from the ATPase domains towards the MCM pore, sequentially engage with DNA (Supplementary Movie 1). This ATPase-controlled sequential movement is reminiscent of the mechanism proposed for 3′-to-5′ single-stranded DNA translocation that underlies replication fork unwinding by the CMG holo-helicase, where neighbouring subunits take sequential steps along DNA resulting in a rotational movement controlled by ATP hydrolysis[37–39]. However, important differences exist. The first difference is technical. The sequential rotary cycling mechanism for single-stranded DNA translocation is commonly accepted because it presents a physically sensible explanation to ATP-hydrolysis powered unidirectional

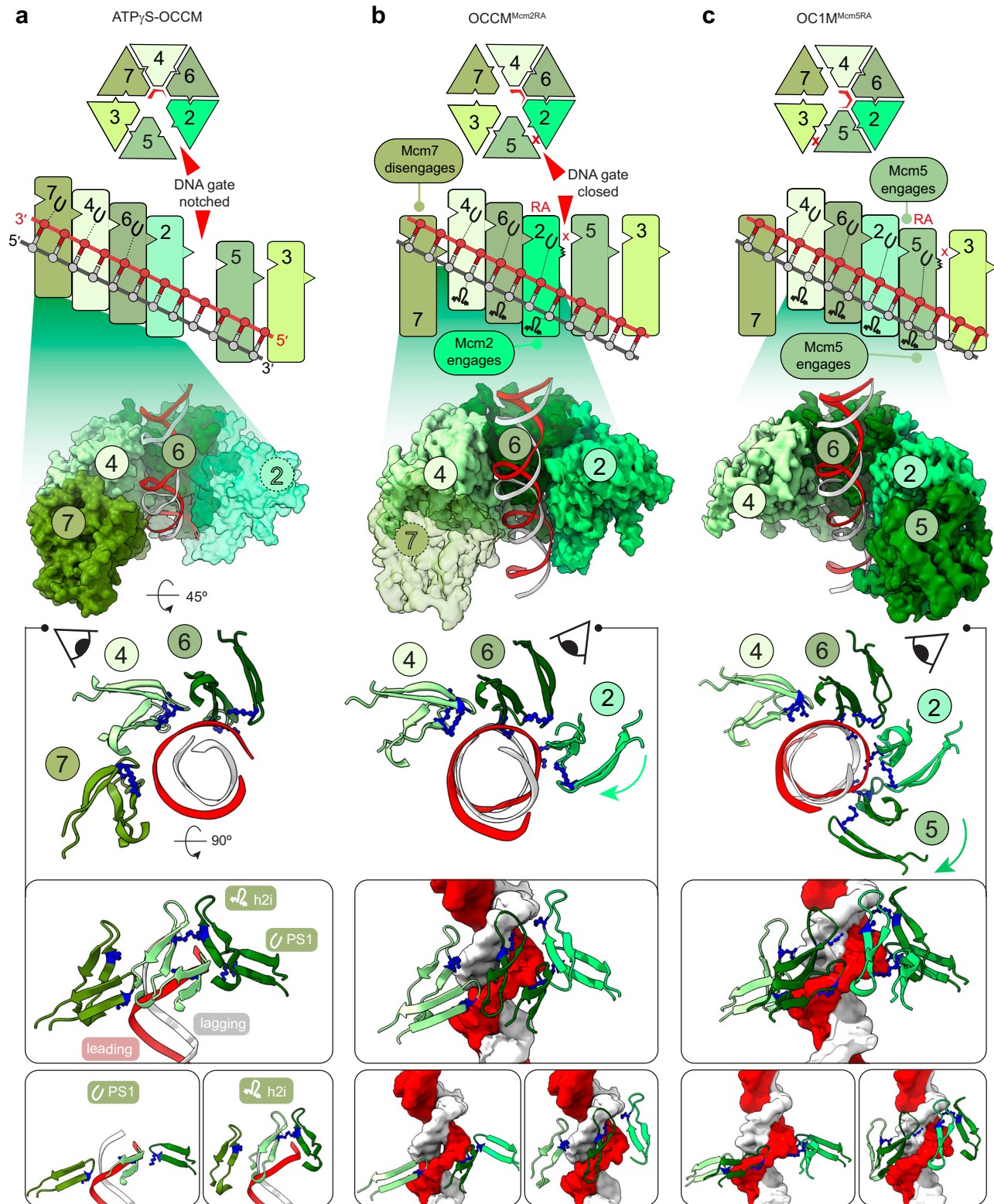

**Fig. 4 | ATPase structure and DNA engagement in ATPγS-OCCM, OCCM^Mcm2RA and OC₁M^Mcm5RA complexes. a** In ATPγS-OCCM (surface representation), Mcm7-4-6-2 ATPases form a right-handed spiral that follows the helicity of DNA. Mcm7, 4 and 6 engage the leading-strand template via a conserved lysine residue of the PS1 hairpin. Mcm2 is not properly engaged with DNA (transparent surface). **b** In OCCM^Mcm2RA (cryo-EM density), Mcm7 (transparent density) disengages from DNA and the right-handed ATPase spiral, which is only formed by Mcm4-6-2. Mcm2 binds DNA. DNA contacts involve PS1 hairpins selectively engaging the leading-strand template and h2i hairpins primarily engaging the lagging-strand template. **c** In OC₁M^Mcm5RA (cryo-EM density), the ATPase domains of Mcm4-6-2-5 form a four-subunit right-handed spiral. Mcm5 engages DNA with the PS1 hairpin binding the leading-strand template and the h2i hairpin binding primarily the lagging-strand template. RA Arginine Finger mutation.

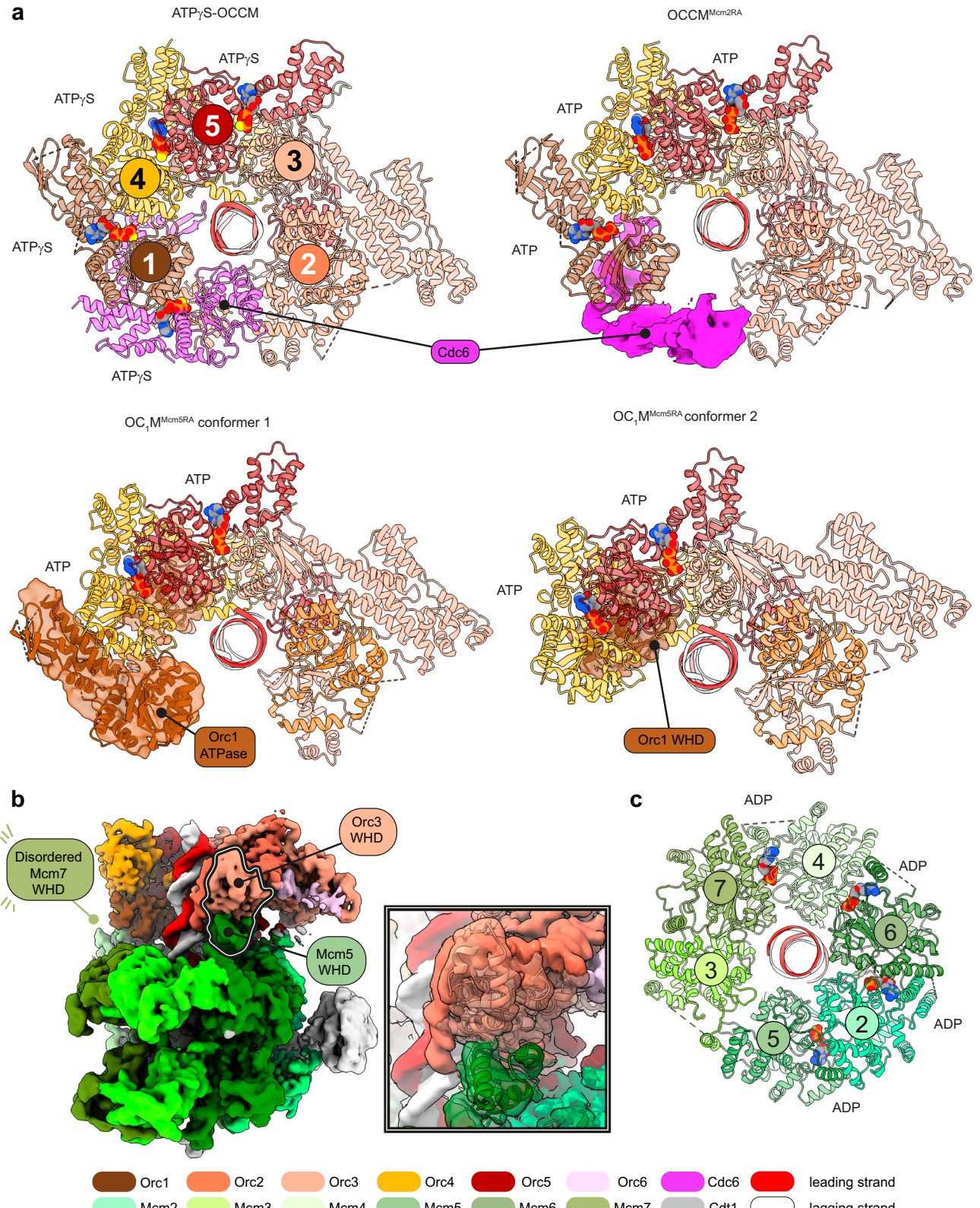

**Fig. 5 | Sequential disassembly of ORC-Cdc6 and structure of ORC-Cdc6/MCM connectivity in OC1M^Mcm5RA. a** In ATPγS-OCCM (top left), the Orc1-Cdc6 and Orc4-Orc1 ATPase sites are ATPγS-engaged and visit a stable form that favours ATP binding. Orc5-4 and Orc3-5 sites are also bound to ATPγS. In OCCM^Mcm2RA (top right), Cdc6 (cryo-EM density) is flexible and ATP can only be observed in the Orc4-Orc1, Orc5-4 and Orc3-5 nucleotide-binding sites. In OC1M^Mcm5RA (bottom), Orc1 (cryo-EM density) is flexible and ATP can only be observed in the Orc5-4 and Orc3-5 sites. **b** Side view of the conformer 2 OC1M^Mcm5RA structure highlights changes in ORC-MCM connectivity compared to OCCM^Mcm2RA. The Mcm7 WHD domain, previously engaged to Orc1, becomes disordered and a new contact is created between the Mcm5 WHD domain and the Orc3 WHD, which could not be detected in OCCM^Mcm2RA. **c** MCM ATPase view shows that the Mcm2-5 gate remains closed in OC1M^Mcm5RA. ADP is bound at four inter-subunit sites.

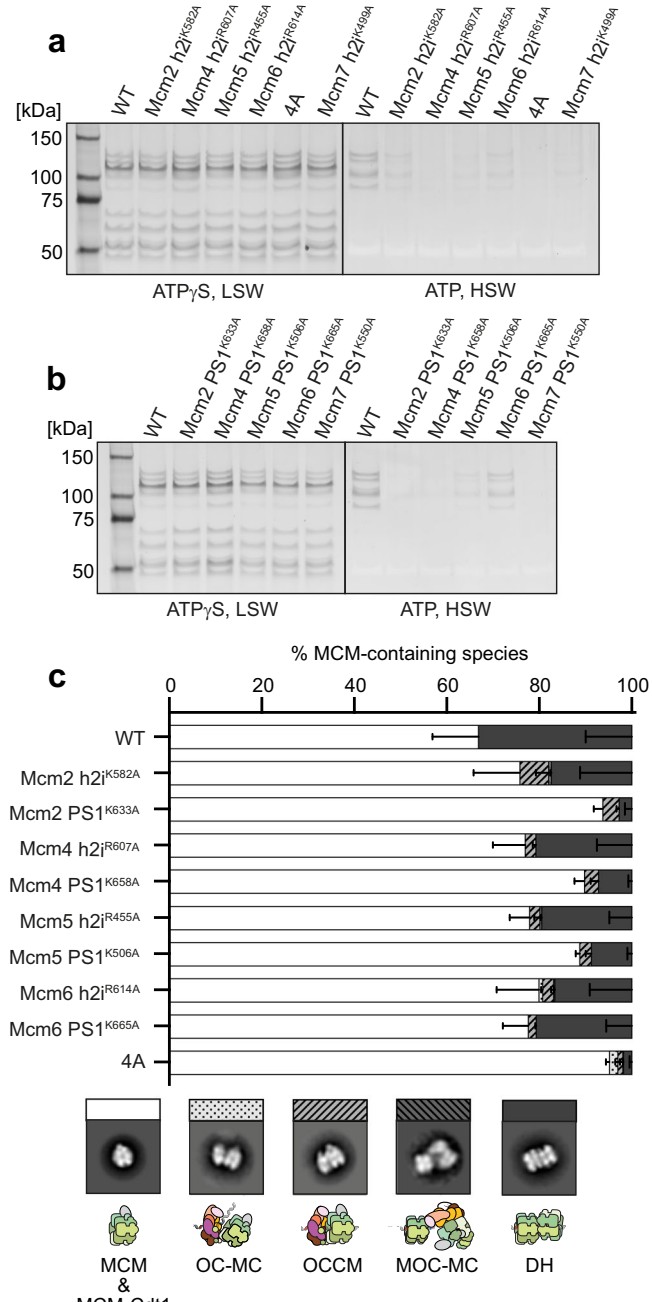

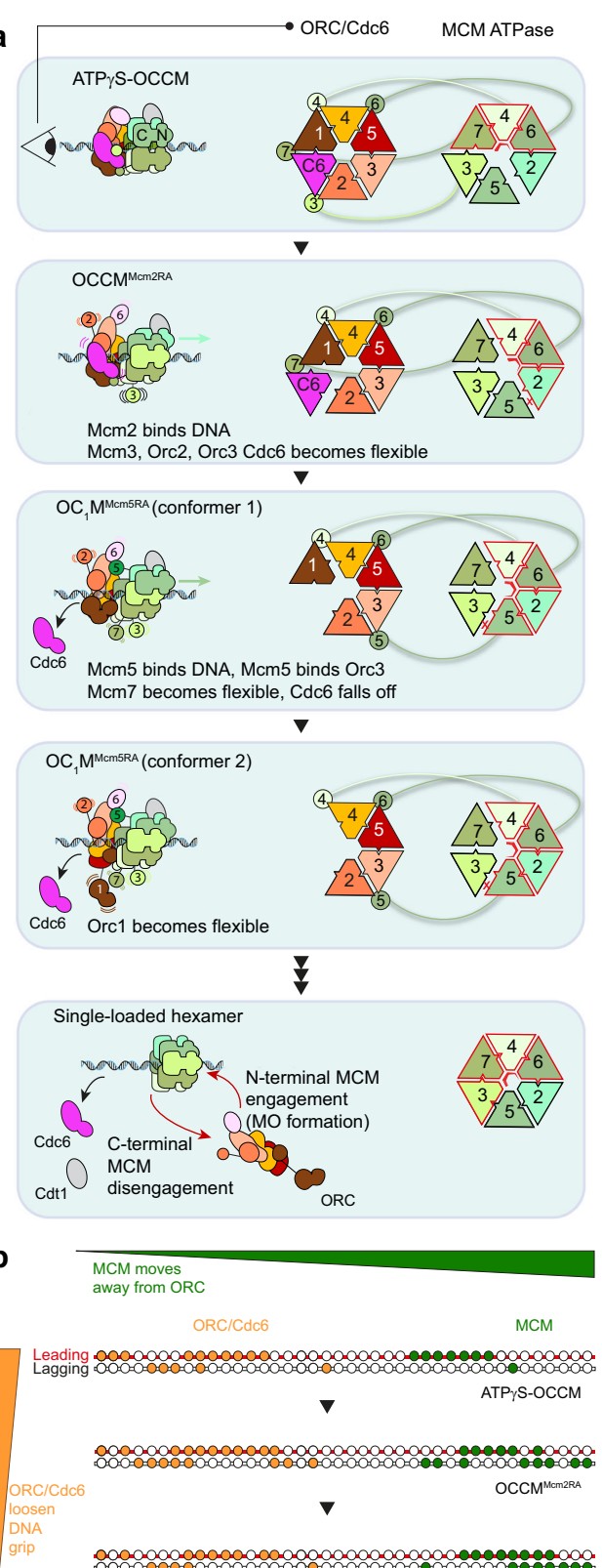

**Fig. 6 | MCM recruitment and DH formation by MCM h2i and PS1 hairpin mutants. a**, **b** Silver-stained SDS-PAGE analysis of MCM recruitment (ATPγS, low salt wash) and DH formation (ATP, high salt wash) using streptavidin-coated magnetic beads and a roadblocked ARS1 origin template for MCM h2i (**a**) and PS1 (**b**) hairpin mutants. This experiment was performed once. **c** Negative stain analysis and 2D averaging of DH loading reactions in solution, with bar graphs showing mean ± SD. N = 3 independent experiments. Source data for all panels are provided as a Source Data file. OC-MC ORC-Cdc6-MCM-Cdt1 complex, OCCM ORC-Cdc6-Cdt1-MCM complex, MOC-MC MCM-ORC-Cdc6-Cdt1 complex, DH MCM double hexamer.

movement[40]. However, to the best of our knowledge, structural studies on DNA unwinding have not yet proven that sequential steps forward taken by neighbouring subunits around the MCM ring (or any NTPase-powered hexameric helicase in general) correlate with translocation away from (or towards) a fixed reference point. In the current study instead, we show that sequential DNA engagement by neighbouring subunits, stabilised using different ATPase variants, correlates with the

distancing of MCM from the ORC loading platform, as well as the weakening of the MCM-ORC interaction interface upon OCCM maturation (Fig. 7b). The second difference is mechanistic. In our model for OCCM-to-SH transition, MCM translocates on duplex DNA with the PS1 hairpin tracking along the leading-strand template with

**Fig. 7 | OCCM maturation to single-loaded hexamer involves duplex DNA translocation by MCM. a** Order of events leading to SH loading onto origin DNA established based on the stepwise disassembly of the ORC/Cdc6 complex, with intermediates stalled by Arginine Finger mutants of MCM. The transition from ATPγS-OCCM to OCCM^Mcm2RA promotes the partial disengagement of Cdc6, Orc2 WHD, Mcm3 WHD and Orc6 from ORC. At the same time, Mcm7 disengages DNA and Mcm2 binds DNA. Transition from OCCM^Mcm2RA to OC₁M^Mcm5RA promotes Cdc6 release, disengagement of the Mcm7 WHD and the stepwise disengagement of the Orc1 ATPase. At the same time Mcm5 binds downstream of the Mcm2 binding site. SH loading upon hydrolysis at the Mcm5-3 ATPase, and possibly other catalytic sites in MCM, promotes Cdt1 release and ORC disengagement from C-terminal MCM. ORC engagement to N-terminal MCM will lead to MO formation and the recruitment of a second MCM hexamer. **b** Schematic view of protein DNA contacts within ATPγS-OCCM, OCCM^Mcm2RA and OC₁M^Mcm5RA shows that ORC-Cdc6 progressively loosens the DNA grip as MCM translocates away from ORC.

the same 3′-to-5′ directionality required for DNA unwinding[13]. The h2i hairpin instead primarily tracks along the lagging-strand template (absent in the fork-unwinding substrate), with 5′-to-3′ polarity. Our observation of double-stranded DNA translocation by MCM is different from the mechanism invoked for the φ29 viral packaging motor, characterised with single-molecule imaging in a classic molecular biology study from Carlos Bustamante's lab. In the φ29 work, backbone phosphate modification was used to show that tracking along one strand only enabled duplex DNA translocation by a ring-shaped ATPase[41]. Conversely, recent structural observations on hexameric RecA and AAA+ ATPases support the notion that duplex DNA translocation can function by tracking along both DNA strands. One is the structure of Ftsk on duplex DNA, where pore elements were observed to engage both leading- and lagging-strand templates[42]. The second is the structure of CMG replicative on duplex DNA at replication termination, where two DNA gripping conformations are observed, which involve contacts with both leading-strand and lagging-strand templates[43].

Based on our data, we propose a structural mechanism whereby OCCM-to-SH transition during MCM loading involves ATP-hydrolysis-driven double-strand translocation of MCM away from ORC. Inspired by published evidence that PS1 hairpin mutations interfere with DNA unwinding[44–46], we have shown that the same point mutations affect loading to varying degrees. Of note, a KA change in the Mcm3 PS1 hairpin, which is lethal in cells, abrogates both the unwinding of an artificial forked DNA substrate[44] as well as loading as observed in our study. Likewise, h2i hairpins establish contacts with the lagging-strand template, which are essential for loading, and in doing so widen the minor groove of the double helix. The same minor-groove widening effect was observed when h2i hairpin contacts are established with the lagging strand, which are essential for nucleating DNA melting upon CMG formation[15]. Collectively, these observations indicate that the MCM motor of the helicase employs the same physical mechanisms of DNA interaction, while it exerts its three functions: loading onto origins, melting of the double helix and unwinding of the established replication fork. What differentiates these three functions are the DNA substrates (duplex or single-stranded DNA) and the ancillary subunits bound to MCM (forming OCCM or CMG complexes).

Several aspects of helicase loading remain to be investigated. For example, how ORC can flip from the C to the N-terminal side of the MCM hexamer upon OCCM to MO transition is not fully understood (Fig. 1a). Single-molecule work indicated that Orc6 functions as the pivot element around which ORC flips[28]. For this mechanism to work, Orc6 would need to stretch extensively, so that it can contact both faces of MCM at the same time. The partial unstructuring of the C-terminal cyclin box of Orc6 observed in our OCCM^Mcm2RA and OC₁M^Mcm5RA might facilitate this transition. Another mysterious aspect of helicase loading pertains to how two single-loaded hexamers can

engage to form a double hexamer. While recent evidence supports a model whereby SHs can meet after moving along DNA by passive diffusion[27], it remains to be established whether their locking into a DH requires ATP hydrolysis. Addressing this question will be of particular interest not only for yeast but also for human MCM loading, as human DH formation is sufficient to nucleate DNA melting in between the two MCM rings, before CMG formation[47,48]. Finally, several studies indicate that ATP hydrolysis by MCM serves to eject Cdt1 during OCCM to SH transition[28,35,36]. It will be important to establish how this function is enacted and how loss of Cdt1 affects MCM's ability to translocate unidirectionally along duplex DNA.

## Methods
### Protein expression and purification
**TrpR.** *Escherichia coli trpR* gene carrying C-terminal Twin-Strep-tag was cloned into a modified expression plasmid based on the 'pET' series (Novagen). The vector was transformed into BL21-CodonPlus (DE3)-RIL competent cells (Agilent cat no. 230245). Cells were grown in AIMTB medium (Foremedium cat no. AIMTB0210), supplemented with 50 μg/mL kanamycin and 34 μg/ml chloramphenicol. Expression was carried out for 48 h at 180 rpm at 37°C. Cells were then centrifuged at 3500 x g for 20 min at 4 °C in a JS-4.2 rotor (Beckman Coulter). Cell pellets were frozen in liquid nitrogen and were stored at -80 °C. Thawed cells were resuspended in lysis buffer (50 mM Tris 8.0, 0.5 M sodium chloride, 2 mM β-mercaptoethanol, 10% glycerol) supplemented with EDTA-free cOmplete protease inhibitor (Roche cat no. 5056489001). The suspension was subjected to sonication for six min at 50% amplitude with a 50% duty cycle. The resulting lysate was centrifuged at 48,000 x g for 30 min in a JA25.50 rotor (Beckman Coulter). The supernatant was incubated with gentle agitation for three h at 4 °C with 2 mL of Strep-Tactin XT 4Flow resin (IBA cat no. 2-5010-025) pre-equilibrated in lysis buffer. The bead slurry was packed into a gravity-flow column and washed with 20 column volumes (CVs) of lysis buffer with protease inhibitor, followed by 20 CV of lysis buffer alone and 20 CV of wash buffer (50 mM Tris 8.0, 0.15 M sodium chloride, 2 mM β-mercaptoethanol). Bound fraction was eluted with elution buffer (1x BXT buffer, IBA cat no. 2-1042-025) and dialyzed overnight against 2 L of dialysis buffer (10 mM Tris 8.0, 0.15 M sodium chloride, 2 mM DTT). The protein solution was concentrated using an Amicon Ultra Centrifugal Filter with 3 kDa MWCO (Millipore cat no. UFC800324) and loaded onto a Superdex 75, HiLoad 16/60 column (Cytiva cat no. 28989333) equilibrated in dialysis buffer. Pure TrpR-containing fractions were pooled, concentrated, aliquoted, flash-frozen in liquid nitrogen, and stored at −80 °C.

**Cdc6.** Expression and purification of *Saccharomyces cerevisiae* Cdc6 was carried out as previously described[20] with some modifications. In brief, BL21(CodonPlus)-RIL cells (Agilent cat no. 230240) were transformed with pAM3 plasmid[20] and were cultured in 6 L of LB medium supplemented with chloramphenicol (34 μg/mL) and ampicillin (100 μg/mL) at 37 °C and 180 rpm until OD₆₀₀ of 0.6 was reached. Protein expression was induced with 0.5 mM IPTG for 5 h at 18 °C. Cells were harvested by centrifugation at 3,500 x g for 20 min using a JS-4.2 rotor (Beckman Coulter). Cell pellets were resuspended in 15 mL of lysis buffer (50 mM potassium phosphate pH 7.6, 5 mM magnesium chloride, 1% Triton X-100, 1 mM DTT, 2 mM ATP, 0.15 M potassium acetate) supplemented with EDTA-free cOmplete protease inhibitor (Roche cat no. 5056489001). The suspension was subjected to sonication for 12 min at 50% amplitude with a 50% duty cycle. The resulting lysate was centrifuged at 48,000 x g for 30 min in a JA-25.50 rotor (Beckman Coulter). The supernatant was incubated with 2.5 mL of pre-equilibrated glutathione sepharose 4B beads (Cytiva cat no.17075601) for 3 h at 4 °C with gentle agitation. The bead slurry was packed into a gravity-flow column and washed with 15 CVs of lysis buffer containing protease inhibitors, followed by 15 CVs of lysis buffer alone. The beads

were resuspended in lysis buffer to form a 50% slurry and incubated with 60 μL of preScission protease (Cytiva cat no. 27084301) overnight at 4 °C with continuous agitation. The flow-through containing the cleaved protein was collected and the concentration of potassium acetate was adjusted to 75 mM with dilution buffer (50 mM potassium phosphate pH 7.6, 5 mM magnesium chloride, 0.1% Triton X-100, 1 mM DTT, 2 mM ATP). Eluate was incubated with 2 mL of Bio-Gel Hydroxyapatite HTP gel (Bio-Rad cat no. 1300150) pre-equilibrated in wash buffer (50 mM potassium phosphate pH 7.6, 5 mM magnesium chloride, 0.1% Triton X-100, 1 mM DTT, 2 mM ATP, 75 mM potassium acetate). The resin was washed with 20 CVs of wash buffer followed by 20 CVs of rinse buffer (50 mM potassium phosphate pH 7.6, 5 mM magnesium chloride, 0.1% Triton X-100, 15% glycerol, 1 mM DTT, 0.15 M potassium acetate). Cdc6 was eluted with elution buffer (50 mM potassium phosphate pH 7.6, 5 mM magnesium chloride, 0.1% Triton X-100, 15% glycerol, 1 mM DTT, 0.4 M potassium acetate). Pooled peak fractions were concentrated using an Amicon Ultra Centrifugal Filter with 3 kDa MWCO (Millipore cat no. UFC800324) and dialyzed twice against 1 L of dialysis buffer (25 mM HEPES pH 7.6, 10 mM magnesium acetate, 0.02% NP-40, 10% glycerol, 5 mM β-mercaptoethanol, 0.1 M potassium acetate), first for 1 h and then overnight. The protein solution was concentrated again, aliquoted, flash-frozen in liquid nitrogen and stored at −80 °C.

**ORC complex.** Expression of *Saccharomyces cerevisiae* ORC complex was carried out as previously described from ySD-ORC strain[20]. Purification of ORC complex was performed as previously described with some modifications. In brief, frozen cell powder was resuspended in lysis buffer (25 mM HEPES pH 7.6, 0.05% NP-40, 10% glycerol, 0.1 M potassium chloride, 2 mM β-mercaptoethanol) supplemented with EDTA-free cOmplete protease inhibitor (Roche cat no. 5056489001). Potassium chloride concentration was adjusted to 0.5 M. The suspension was centrifuged at 235,000 x $g$ for one h at 4 °C using a Ti45 rotor (Beckman Coulter). The supernatant was collected, and calcium chloride was added to a final concentration of 2 mM. The lysate was incubated with calmodulin affinity resin (Agilent cat no. 214303) pre-equilibrated in binding buffer (25 mM HEPES pH 7.6, 0.05% NP-40, 10% glycerol, 0.3 M potassium chloride, 2 mM calcium chloride, 2 mM β-mercaptoethanol) for two h at 4 °C with gentle agitation. The beads were washed with 10 CV of binding buffer and the protein was eluted using elution buffer (25 mM HEPES pH 7.6, 0.05% NP-40, 10% glycerol, 0.3 M potassium chloride, 2 mM EGTA, 1 mM EDTA, 2 mM β-mercaptoethanol). Peak elution fractions were concentrated using a Amicon Ultra Centrifugal Filter with 100 kDa MWCO (Millipore cat no. UFC810024) and further purified by size exclusion chromatography on a Superdex 200, HiLoad 16/60 column (Cytiva cat no. 28989335) equilibrated in gel filtration buffer (25 mM HEPES pH 7.6, 0.05% NP-40, 10% glycerol, 0.15 M potassium chloride, 2 mM β-mercaptoethanol). Pure ORC-containing fractions were pooled, concentrated, aliquoted, flash-frozen in liquid nitrogen, and stored at −80 °C.

**MCM-Cdt1 complex.** Expression of *Saccharomyces cerevisiae* MCM-Cdt1 complexes in yeast was done using strains yAM33 (MCM-Cdt1 WT), yGC216 (Mcm2$^{R676A}$), yGC214 (Mcm3$^{R542A}$), yGC222 (Mcm4$^{R701A}$), yGC226 (Mcm5$^{R549A}$), yGC223 (Mcm6$^{R708A}$), yGC224 (Mcm7$^{R593A}$) and was performed as previously described[29].

Purification of yMCM-Cdt1 complexes expressed in yeast was carried out as previously described[10,29] with some modifications. In brief, thawed cell powder was resuspended in one volume of lysis buffer (45 mM HEPES pH 7.6, 0.1 M potassium acetate, 5 mM magnesium chloride, 0.02% NP-40, 10% glycerol) supplemented with EDTA-free cOmplete protease inhibitor (Roche cat no. 5056489001). The resuspended mixture was clarified by ultracentrifugation at 235,000 x g for one h at 4 °C in a Ti45 rotor (Beckman Coulter). 2 mM calcium chloride was added to the clarified lysate. Lysate was incubated for

three h at 4 °C with 1 mL of calmodulin affinity resin (Agilent cat no. 214303) pre-equilibrated in binding buffer (45 mM HEPES pH 7.6, 0.1 M potassium chloride, 5 mM magnesium chloride, 0.02% NP-40, 10% glycerol, 2 mM calcium chloride). The bead slurry was packed into a gravity flow column and washed with 40 CV of binding buffer. Bound protein was eluted with elution buffer (45 mM HEPES pH 7.6, 0.1 M potassium chloride, 5 mM magnesium chloride, 0.02% NP-40, 10% glycerol, 1 mM EDTA, 2 mM EGTA). The eluate was concentrated using an Amicon Ultra Centrifugal Filter with 100 kDa MWCO (Millipore cat no. UFC810024) and subsequently loaded onto a 120 mL HiLoad Superdex 200 16/60 gel filtration column (Cytiva cat no. 28989335) equilibrated in lysis buffer. Fractions corresponding to the protein of interest were pooled and concentrated again. The concentrated protein was aliquoted, flash-frozen in liquid nitrogen, and stored at −80 °C.

For expression in insect cells (Sf21 cells, Thermo Fisher Scientific cat no. 11497013), wild-type and variant MCM-Cdt1 complexes [(Mcm2$^{K582A}$, Mcm2$^{K633A}$, Mcm3$^{K499A}$, Mcm4$^{R607A}$, Mcm4$^{R658A}$, Mcm5$^{R455A}$, Mcm5$^{K506A}$, Mcm6$^{R614A}$, Mcm6$^{K665A}$, Mcm7$^{K499A}$, Mcm7$^{K550A}$, Mcm2$^{K582A}$-Mcm5$^{R455A}$ [2A], Mcm2$^{K582A}$-Mcm5$^{R455A}$-Mcm6$^{R614A}$ [3A], and Mcm2$^{K582A}$-Mcm$^{R607A}$-Mcm5$^{R455A}$-Mcm6$^{R614A}$ (4A)] carrying an N-terminal 3xFLAG tag with TEV cleavage site on the Mcm3 subunit were cloned into pGBdest vector using the GoldenBac assembly system[49]. Cell cultures were centrifuged at 458 x $g$ for 20 min at 4 °C in a JS-4.2 rotor (Beckman Coulter) and then again at 487 x g for 20 min at 4 °C in an S-4×1000 rotor (Eppendorf). Cell pellets were resuspended in one volume of lysis buffer (45 mM HEPES pH 7.6, 0.1 M potassium chloride, 5 mM magnesium chloride, 0.02% NP-40, 10% glycerol) supplemented with EDTA-free cOmplete protease inhibitor (Roche cat no. 5056489001). The resuspended mixture was clarified by centrifugation at 48,000 x $g$ for 30 min at 4 °C in a JA-25.50 rotor (Beckman Coulter). Lysate was incubated for 3 h at 4 °C with 2 mL of Anti-FLAG M2 affinity gel (Merck cat no. A2220) pre-equilibrated in lysis buffer. The bead slurry was packed into a gravity flow column and washed with 20 CV of lysis buffer supplemented with protease inhibitor, followed by 20 CV or lysis buffer alone. Bound protein was eluted with lysis buffer supplemented with 0.25 mg/mL of FLAG peptide. The eluate was concentrated using an Amicon Ultra Centrifugal Filter with 100 kDa MWCO (Millipore cat no. UFC810024) and subsequently loaded onto a 120 mL HiLoad Superdex 200 16/60 gel filtration column (Cytiva cat no. 28989335) equilibrated in lysis buffer. Fractions corresponding to the stoichiometric complex were pooled and concentrated again. The concentrated protein was aliquoted, flash-frozen in liquid nitrogen, and stored at −80 °C.

**Negative stain analysis**
Negative stain sample preparation was performed on 300-mesh copper grids coated with a carbon film (EM Resolutions cat no. C300Cu). Grids were subjected to glow discharging for 60 seconds at 25 mA (GloQube Plus, Quorum). Four μL of the sample were applied onto the grids and incubated for two minutes. Subsequently, grids were stained twice with 4 μL of 2% (w/v) uranyl acetate, with rapid blotting between applications. The excess stain was removed by blotting the second stain application after 40 seconds. Micrographs were acquired using a Tecnai G2 Spirit TWIN transmission electron microscope (FEI) with a LaB6 electron source operated at an accelerating voltage of 120 keV. Image acquisition was performed with Gatan Digital Micrograph software (v3.53.4137.0) using two cameras: Ultrascan 100XP camera (Gatan) at a nominal magnification of 30,000x (pixel size: 3.45 Å) for MCM Arginine Finger mutant analysis and a RIO 16 camera (Gatan) at a nominal magnification of 21,000x (pixel size: 4.3 Å) for MCM pore loop mutant analysis. Image processing was done using RELION 3.1 or 4.0[50,51]. The contrast transfer function (CTF) of each micrograph was estimated using Gctf v1.06[52]. Initial particle picks were obtained using the autopick function. The selected set of particles was then used as a

training dataset for Topaz v0.2.5[53] picking. Final particle picks were subjected to several rounds of reference-free 2D classification.

## Double hexamer formation reaction in solution for negative stain

**Arginine finger mutants.** Reactions were performed in buffer containing 25 mM HEPES pH 7.6, 100 mM potassium chloride, 10 mM magnesium acetate, and 1 mM DTT. The DNA template used contained an ARS1 origin sequence flanked by HpaII methyltransferase and Widom601-positioned nucleosome at the 5′ end and a Widom603-positioned nucleosome at the 3′ end, MH-NCP-ARS1-NCP[18]. It was used at a concentration of 10 nM and combined with 20 nM ORC, 30 nM Cdc6, and 0.5 mM ATP (final concentrations). The resulting master mix was aliquoted into individual reaction tubes and reactions were initiated by the addition of 20 nM MCM-Cdt1 (final concentration). Loading reactions (final reaction volume 15 μL) were incubated at 30 °C with mixing at 1,250 rpm for 30 min. Samples were immediately subjected to negative staining following the incubation.

**Pore loop mutants.** Reactions were performed in buffer containing 25 mM HEPES pH 7.6, 100 mM potassium chloride, 10 mM magnesium acetate, 1 mM tryptophan and 1 mM DTT. Tryptophan operator sequence-flanked ARS1 DNA template, TrpR-ARS1-TrpR (top 5′-GTT TAAACGATATCCCCGAGAGCATCGAACTAGTTAACTAGTACGCAAGC CGAGATTTTACAGATTTTATGTTTAGATCTTTTATGCTTGCTTTTCAA AAGGCCTGCAGGCAAGTGCACAAACAATACTTAAATAAATACTACTC AGTAATAACCTATTTCTTAGCATTTTTGACGAAATTTGCTATTTTGTA GCTTTGCGTACTAGTTAACTAGTTCGATCGACTCGGGGATATCAAGG TCCGTATACACTAGTTACTCCGCCTAGG-3′) at a concentration of 10 nM was immediately combined with 100 nM TrpR, 20 nM ORC, 30 nM Cdc6, 1 mM ATP and 20 mM MCM-Cdt1 (final concentrations). Loading reactions (final reaction volume 15 μL) were incubated at 30 °C with mixing at 1250 rpm for 30 min. Samples were immediately subjected to negative staining following the incubation.

## Double hexamer formation for pull-down

Reactions were performed in buffer containing 25 mM HEPES pH 7.6, 100 mM potassium chloride, 10 mM magnesium acetate, 1 mM tryptophan and 1 mM DTT, and 0.02% NP-40. TrpR-ARS1-TrpR DNA template at a concentration of 20 nM was immediately combined with 200 nM TrpR, 50 nM ORC, 50 nM Cdc6, 5 (or 315 μM) mM ATP/ATPgS and 220 (or 550) nM MCM-Cdt1 (final concentrations). Loading reactions were incubated at 30 °C with mixing at 1,250 rpm for 30 min. Subsequently, 10 μL of Strep-Tactin type 3 magnetic beads (IBA cat no. 2-1613-002) were added to each 50 μL reaction and incubated at 24 °C for an additional 20 minutes. Unbound components in ATP samples were removed through three washes with 200 μL of high-salt buffer (25 mM HEPES pH 7.6, 1 M sodium chloride, 5 mM magnesium acetate, 1 mM tryptophan, 0.02% NP-40), while unbound components in ATPγS samples were removed by three washes with 200 μL of low-salt buffer (25 mM HEPES pH 7.6, 300 mM sodium acetate, 5 mM magnesium acetate, 1 mM tryptophan, 0.02% NP-40). All samples underwent a final wash with reaction buffer. Washed beads were resuspended in 20 μL of elution buffer (25 mM HEPES 7.6, 100 mM sodium acetate, 25 mM biotin, 10 mM magnesium acetate, 1 mM tryptophan, 0.02% NP-40) and incubated at 24 °C for 10 minutes. Eluted proteins were subjected to SDS-PAGE analysis followed by silver staining (SilverQuest Silver Stain, Invitrogen cat no. LC6070).

## Cryo-EM grid preparation

Reactions were performed in buffer containing 25 mM HEPES pH 7.6, 100 mM potassium chloride, 10 mM magnesium acetate, and 1 mM DTT. MH-capped ARS1 DNA template DNA template (MH-ARS1-MH[18]) at a concentration of 30 nM was combined with 60 nM ORC, 90 nM Cdc6, 60 nM MCM-Cdt1 and either 2 mM (Mcm5RA) or 3 mM

(Mcm2RA) ATP (final concentrations). The resulting loading reactions (total volume: 70 μL) were incubated at 30 °C with mixing at 1250 rpm for 30 minutes. Immediately following incubation, four μL of the reaction mixture was applied to UltrAuFoil R1.2/1.3 or R2/2 grids (Quantifoil cat no. N1-A14nAu30-50 and N1-A16nAu20-01, respectively) coated with a freshly prepared graphene oxide layer[54] for 60 seconds. Grids were subjected to double-sided blotting at a blotting force of 0 for 5 to 6.5 s and plunge-frozen in liquid ethane using a Vitrobot Mark IV (Thermo Fisher Scientific) operating at room temperature and 90% relative humidity.

## Cryo data collection

Data were collected using an in-house Thermo Fisher Scientific Titan Krios transmission electron microscope operated at 300 kV equipped with a Falcon 4i Direct Electron Detector (Thermo Fisher Scientific). Images were acquired automatically using EPU v3.2 software (Thermo Fisher Scientific) in counting mode at 130,000x magnification (0.95 Å per pixel at the specimen level). Three acquisition areas were used per hole. The total electron dose was either 30.34 electrons per Å² or 30.0 electrons per Å² depending on the dataset, with 26 frames or 25 frames respectively. A total of 63,858 micrographs were collected across two separate sessions for the Mcm2RA dataset, while the Mcm5RA dataset comprised 29,191 micrographs collected in one session. Defocus values for all datasets fell within the range of −1.5 to −2.7 μm.

## Cryo data processing

Data processing was conducted using RELION version v3.1 or v4.0[50,51] and cryoSPARC v3 or v4[55]. Motion correction and dose weighting were applied to the raw image data using MotionCor2-1.4.4[56]. Contrast transfer function (CTF) parameter determination for the motion-corrected micrographs using CtfFind4-1.13[57]. Particle picking was performed using Topaz v0.2.5[53]. Density modification was done using DeepEMhancer v0.13[58]

## Mcm5RA

An initial particle set was picked from 400 micrographs using the autopick function in RELION. Iterative rounds of Topaz training and 2D classification within RELION were used to improve particle picking. Subsequently, particles were picked from all 29,191 micrographs, resulting in a total of 1,530,987 particles with a figure of merit threshold of −3. These particles were extracted with a box size of 360 pixels and binned by 2. 2D classification removed contaminant particles, including the MCM-Cdt1 complexes, yielding 672,672 accepted particles. Data processing continued in cryoSPARC with ab-initio model generation followed by 5 rounds of heterogeneous refinement to further clean the dataset. A subset of 314,053 particles resembling the OCCM complex was re-imported into RELION and subjected to 3D classification without alignment and ten classes. Two classes exhibited clear Orc1 ATPase density (totalling 49,788 particles, 15.9%). One class had poor density in the MCM N-tier region (32,955 particles, 10.5%) and was therefore excluded from further analysis. The remaining 231,310 particles (73.6%) were re-extracted unbinned with a box size of 440 pixels and underwent CTF refinement and Bayesian polishing. A subsequent round of 3D classification without alignment with ten classes yielded 8 classes exhibiting low resolution or partial ORC detachment. The remaining two classes, comprising 151,030 particles with well-defined ORC and MCM-Cdt1, were further refined. This particle set was subjected to another round of CTF refinement and Bayesian polishing. Homogeneous and non-uniform refinement in cryoSPARC yielded a 3.4 Å consensus map of the "conformer 2" OC$_1$M$^{Mcm5RA}$ complex. Local refinement of the ORC complex improved resolution to 3.3 Å. To address heterogeneity in the MCM-Cdt1 region, signal subtraction and recentring were performed in RELION followed by 3D classification without alignment. This allowed to separate the particles based on the density of the N-tier of MCM ring and quality of downstream DNA

density. One class comprising 103,314 particles (68.2%) was selected and processed in cryoSPARC to generate the final 3.5 Å map of the MCM-Cdt1 ring. Orc1 density-containing OCCM-like particles were re-extracted unbinned with a box size of 440 pixels and subjected to three rounds of CTF refinement and Bayesian polishing. These particles were subsequently processed in cryoSPARC to produce the final 3.7 Å map of the "conformer 1" OC$_1$M$^{Mcm5RA}$ complex. All final maps were subjected to density modification with DeepEMhancer.

## Mcm2RA

Initial particle picking was performed on 115 micrographs from dataset 1 using Topaz within RELION, employing a model that was used to pick Mcm5RA dataset. Iterative Topaz picking and 2D classification were used to iteratively improve the picking model. Subsequently, particles were picked from all micrographs in both dataset 1 (32,404 micrographs) and dataset 2 (31,454 micrographs), yielding 3,081,872 and 2,249,423 particles, respectively. A figure of merit threshold of -3 was applied, and particles were extracted with a box size of 440 pixels, binned by 4. Multiple rounds of 2D classification were performed separately for each dataset to remove contaminants, including double hexamers and MCM-Cdt1 complexes. The particle sets were then combined and subjected to further 2D classification, resulting in 339,343 OCCM-like particles. Data processing continued in cryoSPARC with ab-initio model generation followed by six rounds of heterogeneous refinement. Homogeneous refinement on the resulting 93,541 particles produced a 3.9 Å reconstruction (particle set I). To increase particle numbers, the Topaz model was retrained on 8000 micrographs from both datasets, leading to 2,330,001 picked particles. The particles were extracted binned by four with 440 pixel box size and subjected to 2D classification rounds, resulting in 539,693 accepted particles. Particles were then re-extracted unbinned. Data processing continued in cryoSPARC with six rounds of heterogeneous refinement. Homogeneous refinement on the resulting 119,406 particles produced a 3.9 Å reconstruction (particle set II). The retraining process was repeated using particle set II as input, generating particle set III (127,420 particles) with a 3.7 Å reconstruction. The three particle sets were combined in RELION (total 340,367 particles) and subjected to 2D classification, resulting in 178,450 particles after duplicate removal. This particle set yielded a 3.5 Å reconstruction. Subsequent 3D classification did not distinguish clear differences in composition or conformation. Thus, all particles were used in one single 3D refinement in cryoSPARC, leading to a 3.3 Å reconstruction. Local refinement improved resolution to 3.2 Å for the ORC and MCM-Cdt1 regions respectively. All final maps were subjected to density modification with DeepEMhancer.

## Model building and refinement

### "Conformer 2" of OC$_1$M$^{Mcm5RA}$ local refinements. MCM-Cdt1 was built to local refinement map (after signal subtraction) density-modified with DeepEMhancer. The MCM-Cdt1 ring from the OCCM structure model (PDB entry 5V8F[22]) was initially fitted as a rigid body into the density using Chimera v1.17.3[59]. Subsequently, the ATPase and N-terminal tiers of the MCM subunits, derived from the phosphorylated DH structure (PDR entry 7P30[60]), were divided into two separate rigid bodies (Mcm2: residues 183-458 and 475-864; Mcm3: 16-339 and 340-738; Mcm4: 177-500 and 501-852; Mcm5: 2-348 and 349-699; Mcm6: 100-463 and 509-842; Mcm7: 3-391 and 392-729) and were superimposed onto the OCCM rigid-body fit and individually fitted into the map in Chimera. The rigid-body docked components were inspected in Coot v0.9.8.1[61], and certain regions were replaced with corresponding segments from AlphaFold predictions[62]. These substitutions were required where the 5V8F model lacked structural information or when the AlphaFold predictions better aligned with the density. Specifically, residues Mcm2 730-757, Mcm4 761-816, Mcm5 440-474, and Mcm6 496-523 were swapped for the AlphaFold

prediction. Conversely, model segments absent from the density but present in the 7P30 model were removed in Coot. The ATPase tiers were then subjected to flexible fitting in Coot for each chain independently. A similar flexible fitting process was applied to the N-terminal tiers, using a blurred density map (blur 50). Additional fragments not supported by the density, which became more apparent after flexible fitting, were deleted. The Cdt1 chain was modelled as two parts (residues 1-437 and 438-604). Residues 1-437 of the AlphaFold prediction were superposed onto the 5V8F structure fit, with residues 366-400 retained from this the 5V8F structure. Residues 438-604 were copied from the 5V8F model as well. Cdt1 segments were subjected to flexible fitting like the MCM subunits. Four ATPase sites were identified as occupied, with ADP modelled based on the density (Mcm5/2, 2/6, 6/4, and 4/7). An alanine residue was introduced at position Mcm5 R549. Subsequently, all chains were merged, and density fit and rotamers were refined manually in Coot. Finally, one round of flexible fitting in ISOLDE v1.7[63] was performed to reduce clashes, requiring the introduction of restraints for cysteines coordinating the zinc atom in the ZnF domains. The model was then subjected to real-space refinement in Phenix v1.21.1[64] against the unmodified map.

ORC complex was built to local refinement map density-modified with DeepEMhancer. The ORC-Cdc6 ring from the OCCM structure model 5V8F was initially fitted as a rigid body into the density in Chimera. Given their apparent absence from the reconstruction, the Cdc6 and Orc1 ATPase domains were deleted, and the remaining ORC complex was again docked as a rigid body. AlphaFold predictions for ORC subunits 1-5 were superimposed to the chain positions of the 5V8F model. Only two helices of Orc6 were visible in the density, and the starting model for this subunit was derived from the 5V8F structure. Subsequent flexible fitting of individual chains and removal of regions lacking density was performed in Coot. Two ATP molecules were identified and built at the interface between Orc subunits 4/5 and 5/3. Unassigned density at the ORC-MCM interface was assigned to the Mcm5 WHD. The AlphaFold model of this domain was rigid-body docked and then flexibly fitted into the density. This assignment was supported by AlphaFold-predicted protein-protein interactions and the domain's positioning, which resembled its conformation in the yCMG structure (PDB entry 7QHS[15]). Subsequently, the model was cross-validated against the 2.5 Å resolution yeast ORC structure (PDB entry 7TJH[65]). Specific regions (Orc2 260-267 and 241-249; Orc3 16-20, 266-275, 610-616; Orc4 475-498; Orc5 412-418) were replaced with corresponding segments from this structure due to better density fit compared to the 5V8F or AlphaFold models. All chains were merged, and density fit and rotamers were manually adjusted in Coot. A final round of flexible fitting in ISOLDE was performed to reduce clashes, followed by real-space refinement in Phenix v1.21.1[64] against the unmodified map.

### OCCM$^{Mcm2RA}$ local refinements. ORC complex was built to local refinement map density-modified with DeepEMhancer. The OC$_1$M$^{Mcm5RA}$ ORC complex model, refined as described above, was fitted into the map in Chimera. As no clear density was observed for Mcm5 WHD, it was deleted from the model. The Orc1 chain was copied from the yeast ORC structure (PDB entry 7TJH) superimposed via Orc4 subunit. The AlphaFold model of Mcm7 WHD was positioned in the density based on its location in the 5V8F model and fitted into the density. The Mcm4 WHD model was replaced with the corresponding segment from the 5V8F structure. These regions were then adjusted as separate chains in Coot through flexible fitting. All chains were subsequently merged, with density fit and rotamer adjustments performed manually in Coot. A final round of flexible fitting in ISOLDE was performed to optimize atomic clashes, followed by real-space refinement in Phenix against the unmodified map.

MCM-Cdt1 was built to local refinement map density-modified with DeepEMhancer. Similarly to ORC complex model, OC$_1$M$^{Mcm5RA}$

MCM-Cdt1 was fitted into the map in Chimera. MCM subunits were split into ATPase and N-terminal tier as described for $OC_1M^{Mcm5RA}$ and were rigid-body docked into the density in Chimera. While minimal adjustments were necessary for Mcm2, 4, and 6, the positioning of Mcm3, 5, and 7 differed quite substantially. The Mcm2 ZnF domain was deleted due to lack of density. The Cdt1 chain was flexibly fitted, and Mcm7 pore loops were added based on the locally refined map after density modification with EMReady v2.0[32]. All six ATPase sites were identified as occupied, with ADP modelled based on the density. An alanine residue was introduced in position Mcm2 R676. Finally, all chains were merged and density fit and rotamers were manually adjusted in *Coot*. Final step was one round of flexible fitting in ISOLDE to reduce clashes (applying restrains for ZnF domains as above). The model was then subjected to real-space refinement in *Phenix* against the unmodified map.

**"Conformer 2" of $OC_1M^{Mcm5RA}$ and $OCCM^{Mcm2RA}$ composite models.** To create the final "conformer 2" $OC_1M^{Mcm5RA\,A}$ and $OCCM^{Mcm2RA}$ models, composite density maps were generated. This involved fitting the locally-refined unmodified maps into the consensus map. All locally refined maps except the MCM-Cdt1 map from the $OC_1M^{Mcm5RA}$ reconstruction (which was recentred following particle subtraction) were already aligned with the consensus map coordinate system. Local maps were then combined using the "vop maximum" function in Chimera. The resulting composite maps were used as input maps for density modification with EMReady[32]. The models refined against the local maps were then fitted into the composite maps, with manual adjustments made to interacting regions in *Coot*. Ideal B-form DNA of random sequence ("GATC" repeat) was then generated in *Coot* and roughly fitted into the "conformer 2" $OC_1M^{Mcm5RA}$ EMReady map. Multiple rounds of flexible fitting, using decreasing distance for self-restraints, were performed to refine the DNA position. The DNA molecule from the "conformer 2" $OC_1M^{Mcm5RA}$ model was copied and flexibly fitted into the consensus EMReady map of $OCCM^{Mcm2RA}$ structure using a similar protocol, although minimal alterations to the DNA position were observed between the two datasets. Components were then merged and subjected to flexible fitting in ISOLDE against the composite map to optimize atomic clashes (applying restrains for ZnF domains as above), followed by real-space refinement in *Phenix* against the unmodified map.

**"Conformer 1" of $OC_1M^{Mcm5RA}$.** ORC complex was built to local refinement map density modified with DeepEMhancer. The "conformer 2" $OC_1M^{Mcm5RA}$ structure was docked into the map in Chimera without requiring adjustments. Orc1 chain was added by superimposing the $OCCM^{Mcm2RA}$ structure onto the model via Orc4 as a reference. Manual density fit adjustments were made in *Coot*. As in "conformer 2", "conformer 1" $OC_1M^{Mcm5RA}$ was found to have ATP bound in two ORC sites. Four MCM ATPase sites were identified as occupied, with ADP modelled as in "conformer 2". Subsequently, all chains were merged, and density fit and rotamers were manually adjusted in *Coot*. A final round of flexible fitting in ISOLDE was performed to reduce clashes (applying restrains for ZnF domains as above), followed by real-space refinement in *Phenix* against the unmodified consensus map.

All refinements were evaluated with MolProbity[66] (Supplementary Tables 1 and 2). Molecular graphics and analyses were performed with UCSF ChimeraX-1.7.1 or UCSF Chimera v1.17.3, developed by the Resource for Biocomputing, Visualization, and Informatics at the University of California, San Francisco[59,67]. All figures show EMReady-generated maps.

**Reporting summary**
Further information on research design is available in the Nature Portfolio Reporting Summary linked to this article.

## Data availability

Cryo-EM density maps have been deposited in the Electron Microscopy Data Bank (EMDB) under the accession codes: EMD-51407 (OCCMMcm2RA composite maps), EMD-51404 (OCCMMcm2RA consensus map), EMD-51405 (OCCMMcm2RA local ORC refinement maps), EMD-51406 (OCCMMcm2RA local MCM-Cdt1 refinement maps), EMD-51441 (OC1MMcm5RA "conformer 1" consensus maps), EMD-51401 (OC1MMcm5RA "conformer 2" composite maps), EMD-51398 (OC1MMcm5RA "conformer 2" consensus map), EMD-51399 (OC1MMcm5RA "conformer 2" local ORC refinement maps), EMD-51400 (OC1MMcm5RA "conformer 2" local MCM-Cdt1 refinement maps). Atomic coordinates have been deposited in the Protein Data Bank (PDB) with the accession codes 9GJW (OCCMMcm2RA), 9GM5 (OC1MMcm5RA "conformer 1"), and 9GJP (OC1MMcm5RA "conformer 2"). The data supporting the findings of this study are available within the paper and its Supplementary Information files. Source data are provided with this paper.

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

## Acknowledgements

We are grateful to the members of the Costa lab, Thomas Miller, Gideon Coster and John Diffley for useful discussion. We thank John Diffley for the gift of yeast expression strains for the MCM-Cdt1 Arginine finger variants. We thank A. Alidoust, N. Patel, and D. Patel in the Structural Biology STP for yeast protein expression. We thank A. Nans, A. Purkiss and J. Cements from the for support with cryo-EM and computing. We thank Oliver Willhoft, Jacob Lewis and Thomas Pühringer for conceiving, cloning and testing the use of Trp repressor as DNA roadblock that stabilises DH loading. This work was supported by the Francis Crick Institute, which receives its core funding from Cancer Research UK (CC2009), the UK Medical Research Council (CC2009), and the Wellcome Trust (CC2009). This work was also funded by the European Research Council (ERC) under the European Union's Horizon 2020 research and innovation programme (grant agreement no. 820102 awarded to A. C.). For the purpose of Open Access, the author has applied a CC BY public copyright licence to any Author Accepted Manuscript version arising from this submission.

## Author contributions

A.B. and A.C. conceived the study. A.B. and J.F.G. produced the MCM-Cdt1 Arginine Finger variants, performed loading reactions and prepared EM and cryo-EM samples. A.B. performed all the work on hairpin variants. A.B. performed all single particle analysis, reconstruction, atomic model building and refinement. A.C. supervised the study. A.B. and A.C. wrote the manuscript.

## Funding

## Competing interests

The authors declare no competing interests.
