## [Transparent Peer Review file · Nature Communications]

Unidirectional MCM translocation away from ORC drives origin licensing

Corresponding Author: Professor Alessandro Costa

Version 0:

Reviewer comments:

Reviewer #1

(Remarks to the Author)

The manuscript by Butryn and colleagues describes analysis of MCM loading, using a set of Arginine finger mutants as well as ps1 and h2i mutants. Their key novel findings are two cryo-EM structures of loading intermediates obtained with the Mcm2RA and Mcm5RA mutants. These intermediates are broadly similar to the previously described OCCM intermediate obtained with WT proteins but with ATPγS, yet harbour specific differences that propose a mechanism for how ATP hydrolysis by MCMs may promote loading.

While the overall data quality is good, I fail to see a major advancement in our understanding of MCM loading. Specifically, while the authors cite suitable literature, they fail to mention and discuss findings that are highly relevant for their own work, and put it all in perspective. Specifically, in PMID:37463200, the Bell lab have studied all six MCM RA mutants and found that sliding of the first MCM ring is impacted. Thus, their results already provide a model for the role of MCM ATPase activity.

While I can appreciate that this work provides structural details - it does so for one very specific stage of MCM loading, and does not relate to other events. For example, there is no mention or discussion of Cdt1 binding/release at all. Several papers suggest that the role of MCM ATPase is to promote ring closure and Cdt1 release (PMID: 28191892, PMID: 25087873), and while these papers are mentioned, these aspects are not discussed or mentioned.

Reviewer #2

(Remarks to the Author)

The manuscript by Butryn, Greiwe, and Costa provides a structure and biochemical investigation of early steps of DNA replication initiation and how the ATPase components of the MCM proteins generate this activity. This is one of the most interesting and fundamental questions of biology, and new structures are extremely important in dissecting the mechanism of this fundamental life process. The authors use mutants of the six MCM ATPase sites to reveal by negative stain EM that two of these mutants accumulate a specific intermediate species along the pathway towards formation of a double-hexamer. The paper determines the full structures for both mutants by cryo-EM to find differences between the two- along with that of an earlier intermediate structure- that provides a set of subtly different states that the authors place in mechanistic sequence. The manuscript doesn't answer everything, but this is an important step forward in understanding a stepwise process of DNA replication initiation. The structures appear well-executed, but some of the writing and figures could be improved. The following comments may assist the manuscript's clarity for a general audience.

Lines 54-55: "concerted sequential"- these cannot both be true. If steps occur sequentially, they are not concerted.

Line 104: "we could also observe all DH loading intermediates"—it is not known to be true that ALL are observed. The point here is that none have been specifically removed. The things that are observed are those that are sufficiently well-defined to be identified during EM 2D classification. This may or may not encompass 100% of loading intermediates. The word "all" is too strong.

Although the Mcm2 and Mcm5 are adjacent in the ring, only the Mcm2 arginine finger is involved at the Mcm2/5 "gate" interface. The description of both arginine fingers as gate subunits (for example at line 114) is somewhat misleading, but not

technically wrong as written. It may be good to explicitly define the attributes that comprise the Mcm2/5 ATPase site.

Fig. 2- a schematic panel of the order of MCM subunits around the ring that also defines the subunit that contributes Walker-A/B motifs and the subunit that contributes the arginine finger would be very helpful. A schematic panel is present in Fig. 4 that could be used. Please also indicate the attributes that define the different ATPase motifs and whether the view is from the “top” or “bottom” of the ring.

Fig. 2- I have trouble distinguishing the grey shades used. Could these be with different colors?

Although the schemes at the top of Fig. 4 are easy to interpret, the rest of the figure is very difficult to understand. It is confusing which hairpins correspond to h2i and ps1b.

The authors have placed 3 similar structures in a defined order within figure 7, and it seems to me that this is ultimately based on some reasoning that should be provided in the discussion.

I have a question after looking at Fig. 7. The illustrated RA mutants disrupt an interface within the sequence of the illustrated pathway and provide an intersubunit “gap” that would close in subsequent steps. Within the illustrated pathway, the arginine fingers of Mcm4 and Mcm6 are engaged to their neighbor at all steps of the sequence. Would this lead to the prediction that 4RA and 6RA should be most prone to stopping at the first species and not lead to the intermediate at all? This is largely observed in Fig. 2 for 6RA but not 4RA.

Reviewer #3

(Remarks to the Author)

This work addresses the role of Mcm2-7 ATP hydrolysis in the loading of the first MCM ring by ORC-Cdc6, leading to the separation of ORC from the first loaded Mcm2-7 hexamer. The authors identified by negative stain EM that the arginine fingers of Mcm2 and Mcm5 are important for this process and went on to determine cryo-EM structures of these two mutant OCCM complexes (Mcm2RA and Mcm5RA). By comparing these new structures with the ATPgS bound WT OCCM structure reported previously by H Li lab, they observed separation of Cdc6 from ORC-Cdc6, modified DNA engagement of the Mcm2-7 subunits, as well as the gradual movement of MCM away from ORC. These observations have enabled them to propose a mechanism in which Mcm2-7 migrates away from ARS1 bound ORC to complete the loading of the first MCM and perhaps in preparation of the recruitment of the second MCM. The structural analyses were carefully done and clearly described, the associated mutagenesis and in vitro assays complement structural observations, the figures are well illustrated, and the literatures – in my opinion – are properly cited. So, the manuscript was a pleasure to read, and this work indeed advances our understanding of the origin activation, a key event in central dogma. We only have a few minor suggestions and typo corrections.

1. The title is a bit confusing. I think it is MCM that translocates away from ORC, not DNA translocates itself.
2. Line 88, specify Arg numbers in the RA mutations.
3. Some figure panels may have been cited out of order. Line 169-172, line 184-206, Figs. 5a > 3b > 5 > 4 > 5; line 177-188, SI Fig. 4 and 5; Line 238-248, SI Fig. 8 > Fig. 7e).
4. Line 124, need to use consistent abbreviations. OCCMMcm2RA is different from OCCM2RA in SI Fig. 2 and 3 legends, and OC1MMcm5RA (line 180) is different from OC1M5RA in SI Fig. 4 and 5 legends.
5. Line 131-132, “Within ORC-Cdc6 ... in OCCMMcm2RA”, sounds redundant.
6. Line 624-687, change the order of presentation of OCCMMcm2RA and OC1MMcm5RA data processing as well as model building and refinement, to be consistent with main text.
7. OCCMMcm2RA is sometimes referred to OCCMMcm2RA. Same for OC1M5RA.
8. SI Fig. 1b, NUC, nucleosome?
9. SI Fig. 2 legend, “Mcm2 RA mutant”: “Mcm2RA mutant” (as in SI Fig. 1c-h)?
10. SI Fig. 2c-f and 3 are cited out of order, so are SI Fig. 4c-f and 5.
11. In methods, define yCdc6 with the strain? Also, for other proteins.
12. Line 52, use prime symbol not apostrophe (3'), also check other places.
13. Line 168, Orc3-Orc4 should be Orc3-Orc5? Also line 190.
14. Line 547, μ L but not uL, also check other places, e.g., line 563, 572.
15. Line 600, should be “90% relative humidity”.
16. SI Fig. 2 legend line 5, (c) should be (d)?
17. SI Fig. 3, upper left, 2x 2D CI should be “Class”?

Reviewer #4

(Remarks to the Author)

Reviewer #1 (Remarks to the Author):

While the overall data quality is good, I fail to see a major advancement in our understanding of MCM loading. Specifically, while the authors cite suitable literature, they fail to mention and discuss findings that are highly relevant for their own work, and put it all in perspective.

We would like to thank the reviewer for the assessment of our manuscript and for acknowledging the quality of our work. The points raised invite us to expand the description of single molecule fluorescence work on MCM loading, which we have now done in the Results and in the Discussion section.

Specifically, in PMID:37463200, the Bell lab have studied all six MCM RA mutants and found that sliding of the first MCM ring is impacted. Thus, their results already provide a model for the role of MCM ATPase activity.

We now mention these findings as indicated by the reviewer when stating “We note that the order of events derived from our structures agrees with single-molecule co-localisation observations of MCM loading, performed with wild-type or Arg Finger mutant proteins^{28,35,36}. These studies previously implicated ATP hydrolysis by MCM in facilitating sliding along DNA”. It should be noted that Bell and colleagues describe sliding but do not discriminate between passive diffusion and ATPase driven unidirectional translocation, which is the point of our paper.

While I can appreciate that this work provides structural details - it does so for one very specific stage of MCM loading, and does not relate to other events.

We are in complete agreement with the reviewer. Our study is focused on describing the effect of ATPase mutants that completely (or almost completely) block MCM loading. We focus on one specific stage of MCM loading, OCCM maturation, which we think is worth studying because it is essential.

For example, there is no mention or discussion of Cdt1 binding/release at all.

We did not mention Cdt1 release because our structural intermediates do not address this step. The reviewer is right in stating that this is an important aspect of MCM loading. Our discussion session now ends with the sentence: “several studies indicate that ATP hydrolysis by MCM serves to eject Cdt1 during OCCM to SH transition^{28,35,48}. Establishing how this function is enacted and how loss of Cdt1 affects MCM’s ability to translocate unidirectionally along duplex DNA will be important.”

Several papers suggest that the role of MCM ATPase is to promote ring closure and Cdt1 release (PMID: 28191892, PMID: 25087873), and while these papers are mentioned, these aspects are not discussed or mentioned.

Cdt1 release is addressed in the point above. Regarding Mcm2-5 gate closure, we have now edited the Discussion section as follows: “These [single molecule fluorescence] studies previously [...] showed that ORC release occurs after closure of the Mcm2-5 gate, in

accord with our structural observations.”

Reviewer #2 (Remarks to the Author):

We thank the reviewer for recognising that understanding the role of ATP hydrolysis by MCM in replication initiation is “one of the most interesting and fundamental questions in biology”. Also, we thank for recognising the important role of Structural Biology in understanding “the mechanism of this fundamental life process”. Finally, we are grateful for recognising that the work is “well executed” and that our manuscript “is an important step forward in understanding a stepwise process of DNA replication initiation.”

Lines 54-55: “concerted sequential”- these cannot both be true. If steps occur sequentially, they are not concerted.

Now changed to “sequential”.

Line 104: “we could also observe all DH loading intermediates”—it is not known to be true that ALL are observed. The point here is that none have been specifically removed. The things that are observed are those that are sufficiently well-defined to be identified during EM 2D classification. This may or may not encompass 100% of loading intermediates. The word “all” is too strong.

We changed “all” to “all known”. Thank you for the remark.

Although the Mcm2 and Mcm5 are adjacent in the ring, only the Mcm2 arginine finger is involved at the Mcm2/5 “gate” interface. The description of both arginine fingers as gate subunits (for example at line 114) is somewhat misleading, but not technically wrong as written. It may be good to explicitly define the attributes that comprise the Mcm2/5 ATPase site.

We respectfully disagree with this point for the reviewer. It is correct that Mcm2 and Mcm5 are DNA-gate subunits and we are claiming that Arg Finger mutations in these two subunits block MCM maturation. We are not implying that the Arg finger of Mcm5 faces the DNA gate.

Fig. 2- a schematic panel of the order of MCM subunits around the ring that also defines the subunit that contributes Walker-A/B motifs and the subunit that contributes the arginine finger would be very helpful. A schematic panel is present in Fig. 4 that could be used. Please also indicate the attributes that define the different ATPase motifs and whether the view is from the “top” or “bottom” of the ring.

Thank you for this helpful comment. We added this schematic panel in Figure 2 as suggested. This panel additionally provides a visual representation of the relative impact of specific RA mutations on the efficiency of DH formation around the MCM ring.

Fig. 2- I have trouble distinguishing the grey shades used. Could these be with different colors?

We included patterns to improve graph bar interpretability.

Although the schemes at the top of Fig. 4 are easy to interpret, the rest of the figure is very difficult to understand. It is confusing which hairpins correspond to h2i and ps1b.

Fair point. We revised Figure 4 by swapping and rotating the bottom panels, and by including additional panels that highlight the individual interactions between the PS1 and h2i loops with DNA. We believe these modifications enhance the clarity of the figure.

The authors have placed 3 similar structures in a defined order within figure 7, and it seems to me that this is ultimately based on some reasoning that should be provided in the discussion.

We have revised the discussion to better explain the logic behind the ordering of the three structural intermediates.

I have a question after looking at Fig. 7. The illustrated RA mutants disrupt an interface within the sequence of the illustrated pathway and provide an intersubunit “gap” that would close in subsequent steps. Within the illustrated pathway, the arginine fingers of Mcm4 and Mcm6 are engaged to their neighbor at all steps of the sequence. Would this lead to the prediction that 4RA and 6RA should be most prone to stopping at the first species and not lead to the intermediate at all? This is largely observed in Fig. 2 for 6RA but not 4RA.

The reviewer brings up a valid point. In comparing the effect of Mcm4RA and Mcm6RA in yeast MCM loading and in translocation by the *Drosophila* CMG (Ilves et al Mol Cell 2010), we note that the effect of Mcm4RA and Mcm6RA give similar defects in the two assays. For the translocation mechanism this has been rationalised with the speculation that different ATPase sites might tolerate inactivation that only mildly disrupt the sequential rotary cycling mechanism. Something similar might be true for OCCM maturation to SH, however we prefer to refrain from commenting on this aspect, due to the lack of structures for the Mcm6 and Mcm4RA variants.

Reviewer #3 (Remarks to the Author):

We would like to thank this reviewer for stating that “The structural analyses were carefully done and clearly described”, the figures “well illustrated” and the literature “properly cited”. It was nice to read that the manuscript was deemed “a pleasure to read” and that our work, in the reviewer’s opinion “advances our understanding of the origin activation, a key event in central dogma”.

1. The title is a bit confusing. I think it is MCM that translocates away from ORC, not DNA translocates itself.

We changed the title to "Unidirectional MCM translocation away from ORC drives origin licensing"

2. Line 88, specify Arg numbers in the RA mutations.

Done.

3. Some figure panels may have been cited out of order. Line 169-172, line 184-206, Figs. 5a > 3b > 5 > 4 > 5; line 177-188, SI Fig. 4 and 5; Line 238-248, SI Fig. 8 > Fig. 7e).

We recognise that this is an issue, however we believe that changing the order will make figures hard to understand. For all main or supplementary figures, figures are referenced sequentially and all panels within a figure are cited in order. Panels in all figures are mainly grouped based on the methodology, which poses restrictions on referencing order. It is however key that results for different mutants are shown side-by-side for the reader to appreciate e.g. structural differences. This is how, for example, the following order has been generated: Figure 4a > 4b > 5a > 3b (referenced again) > 5a (referenced again) > 5b > 4b (referenced again) > 4c > 5c > 4c (referenced again).

4. Line 124, need to use consistent abbreviations. OCCMMcm2RA is different from OCCM2RA in SI Fig. 2 and 3 legends, and OC1MMcm5RA (line 180) is different from OC1M5RA in SI Fig. 4 and 5 legends.

Thank you for pointing this out. All inconsistent abbreviations have now been fixed.

5. Line 131-132, "Within ORC-Cdc6 ... in OCCMMcm2RA", sounds redundant.

This sentence has now been revised.

6. Line 624-687, change the order of presentation of OCCMMcm2RA and OC1MMcm5RA data processing as well as model building and refinement, to be consistent with main text.

The order of presentation of data reflects the chronological order in which the cryo datasets were processed and the structures built. For example, picking model derived from the Mcm5RA datasets was subsequently used to pick the particles from the Mcm2RA dataset. Similarly, OC₁M^{Mcm5RA} conformer 2 structure was built as first, and this model was then used for the refinements of other models. We maintained the original order because we think that it will help the readers to understand the subsequent steps that have been taken.

7. OCCMMcm2RA is sometimes referred to OCCMMcm2RA. Same for OC1M5RA.

All inconsistent abbreviations have now been corrected.

8. SI Fig. 1b, NUC, nucleosome?

Yes. We have added all abbreviations to SI Fig 1.

9. SI Fig. 2 legend, "Mcm2 RA mutant": "Mcm2RA mutant" (as in SI Fig. 1c-h)?

All inconsistent abbreviations have now been corrected.

10. SI Fig. 2c-f and 3 are cited out of order, so are SI Fig. 4c-f and 5.
Please see answer to point 3.

11. In methods, define yCdc6 with the strain? Also, for other proteins.
Saccharomyces cerevisiae Cdc6 was expressed in *Escherichia coli* cells, not in yeast.
We revised this part of materials and methods to make that clearer for all listed proteins. All used yeast strains are mentioned in the text with citations.

12. Line 52, use prime symbol not apostrophe (3'), also check other places.
This has been fixed for all instances in the text, Figure 4 and Supplementary Figure 6.

13. Line 168, Orc3-Orc4 should be Orc3-Orc5? Also line 190.
This has been fixed. Thank you for spotting this.

14. Line 547, μL but not uL, also check other places, e.g., line 563, 572.
This has been fixed.

15. Line 600, should be "90% relative humidity".
This has been fixed.

16. SI Fig. 2 legend line 5, (c) should be (d)?
This has been fixed.

17. SI Fig. 3, upper left, 2x 2D Cl should be "Class"?
This and other inconsistencies in Supplementary Figure 3 have been fixed.

Reviewer #4 (Remarks to the Author):

We would like to thank this reviewer for their contribution.